

**Multistatic meteor radar observations of two-dimensional horizontal MLT wind**
Wen Yi [1,2], Jie Zeng[1,2], Xianghui Xue [1,2,3*], Iain Reid[4,5], Wei Zhong[1,2], Jianfei
Wu[1,2], Tingdi Chen[1,2], Xiankang Dou[1,6]
[1] Deep Space Exploration Laboratory / School of Earth and Space Sciences, University
of Science and Technology of China, Hefei 230026, China
[2] CAS Key Laboratory of Geospace Environment/CAS Center for Excellence in
Comparative Planetology, Anhui Mengcheng Geophysics National Observation and
Research Station, University of Science and Technology of China, Hefei 230026, China
[3] Collaborative Innovation Center of Astronautical Science and Technology, Harbin,
China
[4] ATRAD Pty Ltd., Adelaide, SA 5000, Australia
[5] School of Physical Sciences, University of Adelaide, Adelaide, SA 5000, Australia
[6] Electronic Information School, Wuhan University, Wuhan, China
**Corresponding author:** Xianghui Xue (xuexh@ustc.edu.cn)
**Abstract:** All-sky meteor radars have become a reliable and widely used tool to observe
horizontal winds in the mesosphere and lower thermosphere (MLT) region. The
horizontal winds estimated by conventional single-station radars are obtained after
averaging all meteor detections based on the assumption of the homogeneity of the
horizontal wind in the meteor detection area (approximately 200-300 km radius). In this
study, to improve the horizontal winds, we apply a multistatic meteor radar system
consisting of a monostatic meteor radar in Mengcheng (33.36 °N, 116.49 °E) and a
bistatic remote receiver in Changfeng (31.98 °N, 117.22 °E), separated by
approximately 167 km to increase the number of meteors by at least 70% and provide
two different viewing angles of the meteor echoes. The accuracy of the horizontal wind
measurement depends on the meteor number in time and altitude intervals. Compared
to typical monostatic meteor radar, our approach shows the feasibility of estimating the
two-dimensional horizontal wind field. The technique allows us to estimate the mean
horizontal wind and the gradient terms of the horizontal wind, moreover, the horizontal
divergence, relative vorticity, stretching and shearing deformation of the wind field. We
are confident that the improved horizontal wind parameters will contribute to improving
the understanding of the dynamics in the MLT region.
Keywords: Meteor radar, mesosphere and lower thermosphere region, horizontal wind



## Introduction

The mesosphere and lower thermosphere (MLT) are important connecting regions that couple the lower and upper atmosphere through a variety of atmospheric waves, such as gravity waves, tides and planetary waves (e.g., Salby, 1984, Fritts and Alexander, 2003; Forbes and Garrett, 1979). Observation of these atmospheric dynamical processes is very important for understanding the coupling between atmospheric layers. In recent decades, significant development of ground-based techniques, such as radars and LIDARs, have permitted observations of MLT dynamics at different spatial and temporal scales, as well as their long-term climatology from the equator to the poles. In particular, meteor radar has become the most widely used instrument to routinely observe MLT winds among ground-based techniques because it has the advantages of being low cost and easy to install, and operate automatically and continuously under all weather conditions (e.g., Hocking et., 2001; Holdsworth et al., 2004; Fritts et al., 2010; Yu et al., 2013; Jia et al., 2018; Spargo et al., 2019; Yi et al., 2019; 2021).

The radar technique for atmospheric wind measurement by detecting the radial drift velocity of the meteor ionized trial began in the 1950s, and within a few decades, pioneering studies of the mesospheric dynamics and their climatology were conducted (e.g., Robertson et al., 1953; Elford and Robertson, 1953; Elford, 1959; Roper and Elford, 1963; Roper, 1966, 1975; and see Reid and Younger (2016) for a brief history of these early observations). From the late 1970s to the 1980s, the applications of meteor radar in mesosphere wind research diminished along with the retirement of some active researchers and important facilities; moreover, another radar technique using partial reflection operated in medium/high frequencies (M/HF) became more common (e.g., Reid, 2015). At the same time, a few interesting experiments using meteor trails to measure upper atmosphere winds were pioneered on the spaced antenna array of VHF (Very High Frequency) MST (Stratosphere, Troposphere, Mesosphere) and ST Doppler radars (e.g., Aso et al., 1979; Avery et al., 1983; Tsuda et al., 1987; Cervera and Reid, 1995; Hocking, 2011, and the references therein). However, all these early meteor radars suffer from low meteor detection rates. In the 1990s, the rebirth of





meteor wind radar, also called all-sky meteor radar, was made possible by the development of inexpensive personal computers, solid-state radar transmitters and better data acquisition systems, as well as the interferometric technique of antenna configuration (e.g., Jones et al., 1998). In the 21$^{st}$ century, the meteor radar has become a standard tool for the routine measurement of the horizontal wind and dynamics in MLT and largely displaced the previous instruments with similar functions, such as partial reflection radars (e.g., Vincent and Reid, 1983, Reid, 2015), ISR radar (e.g., Nicolls et al., 2010) and VHF Doppler radar (e.g., Reid et al., 1988).

A typical all-sky meteor radar consists of a pair of crossed dipoles (e.g., Hocking et al., 2001; Holdsworth et al., 2004) or a group of a few transmitting elements for transmission (e.g., Fritts et al., 2010; 2012) collocated with five pairs of crossed dipoles arranged in a cross as an interferometric receiving antenna array (e.g., Jones et al., 1998). This configuration is also called monostatic or single-station meteor radar and observes the backscatter meteor echo. The winds are estimated by monostatic meteor radar assuming that the horizontal wind is homogeneous inside the meteor detection volume (approximately 200–300 km radius). The wind measurements normally have spatial and temporal bins of 2-3 km and 0.5-2 hours in the approximate altitude range of 70–110 km, respectively. These measurements have made significant contributions to understanding the behaviour of large-scale atmospheric waves, such as planetary waves (e.g., Vincent, 2015 and the references therein) and tides (e.g., Manson et al., 2002; Jacobi, 2012; Stober et al., 2021a) in the MLT region. In addition, although there is some controversy concerning the accuracy and composite temporal window (e.g., Vincent et al., 2010; Fritts et al., 2012), monostatic meteor radars have been developed to estimate gravity wave momentum fluxes because of substantial continuous data all over the world (e.g., Hocking et al., 2005; Fritts et al., 2010., Andrioli et al., 2013; Jia et al., 2018, and references therein).

In addition to the now dominant all-sky monostatic (backscatter) meteor radar, early meteor radars were designed as multi-station systems using forward scattering meteor echoes. This was because these early radars operated as continuous wave radars,



requiring separation between the transmitter and receivers. For example, a famous
meteor radar was built to measure the upper atmosphere wind in 1958 at the St Kilda
site, near Adelaide. This radar system consisted of a transmitting station and a remote
receiving system approximately 23 km from the transmitter, and the receiving system
had a main site and two supplementary receiving sites approximately 5 km north and
east (Roper and Elford, 1963; Roper, 1966). A similar meteor radar system with a 27
km distance between the transmitting station and receiving station was installed in
Atlanta, GA, USA (e.g., Roper, 1975). Since then, however, this type of radar has
gradually been replaced by monostatic narrow beam (e.g., Cervera and Reid, 1995) and
then all-sky (e.g., Hocking et al., 2001; Holdsworth et al., 2004) radars for measuring
MLT region dynamics. Recently, some innovative multistatic meteor radar systems,
such as the MMARIA (multistatic and multifrequency agile radar for investigations of
the atmosphere) (Stober and Chau, 2015; Stober et al., 2018), SIMO (single-input
multiple-output) (Spargo et al., 2019), and SIMONe (Spread Spectrum Interferometric
Multistatic meteor radar Observing Network) (Conte et al., 2021; Chau et al., 2021),
have been designed and proven to increase the number of meteor detections and the
diversity of viewing velocity angles. Thus, multistatic meteor radar systems have
several advantages over classic monostatic meteor radars, such as obtaining higher-
order wind field information (e.g., Stober et al., 2015; 2018, Chau et al., 2017), vertical
velocity (e.g., Chau et al., 2021; Stober et a., 2022) and mesoscale dynamics (e.g.,
Spargo et al., 2019; Conte et al., 2021; Volz et al., 2021; Stober et al., 2021b).
This study describes a multistatic meteor radar system consisting of a monostatic
meteor radar and a bistatic remote receiver separated by 167 km and presents
preliminary results of the derived two-dimensional wind fields in the MLT region over
Central-Eastern China. Our paper is organized as follows. In section 2, we present the
experimental instruments and their arrangement. Then, in section 3, we introduce the
measurements obtained by the radar system. The preliminary results of improved wind
estimations are presented in section 4. Finally, we summarize our results and discuss
multistatic meteor radar system expansion in the future.



## 2 Instrumentation and Data

The multistatic meteor radar considered in this study consists of a meteor radar located at Mengcheng (33.4 N, 116.3° E) and a remote receiving system located at Changfeng (31.98° N, 117.22° E) in Hefei city, Anhui Province. Figure 1 shows the schematic diagram of the backward and forward scatter geometry for the Mengcheng meteor radar and the Changfeng remote receiver, hereafter MCR and CFR. The Changfeng remote receiver is located southeast of Mengcheng, and the distance between the two sites is approximately 167.6 km.

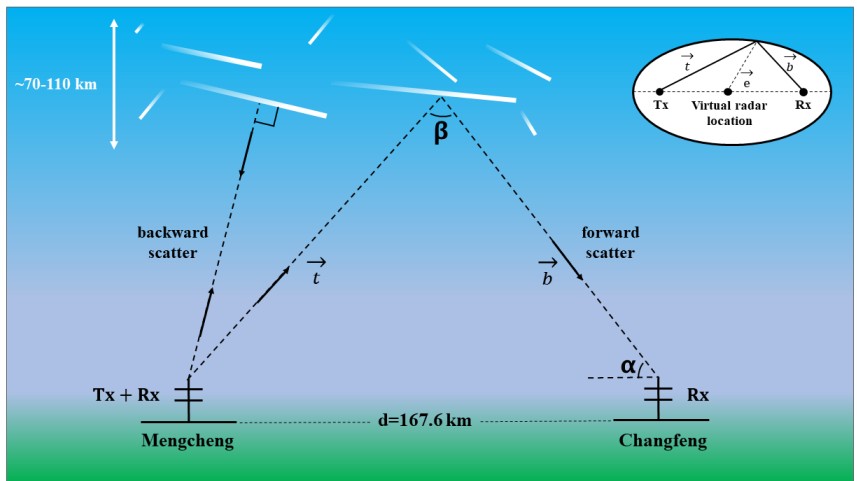

**Figure 1.** Schematic diagram of a backward scatter and forward scatter geometry for Mengcheng meteor radar and Changfeng remote receiver.

**Table 1.** Main operation parameters of the Mengcheng meteor radar

| Frequency | 38.9 MHz |
|---|---|
| Peak power | 20 kW |
| Pulse repetition frequency (PRF) | 430 Hz |
| Coherent integrations | 4 |
| Range resolution | 1.8 km |
| Pulse type | Gaussian |
| Pulse code | 4 bit complementary |
| Pulse width | 24µs |
| Duty cycle | 15% |
| Detection range | 70-110 km |





The Mengcheng all-sky meteor radar (MCR) has been operating since April 2014, at a
frequency of 38.9 MHz, and a peak power of 20 kW. The Mengcheng meteor radar
belongs to the meteor detection radar (MDR) series manufactured by ATRAD and is
similar to the Buckland Park meteor radar system described by Holdsworth et al. (2004).
A single crossed and folded dipole is used for transmission. Five two-element Yagi
antennas using a cross '+' shape arrangement (Jones et al., 1998) are used for reception.
Table 1 shows the experimental parameters used for the Mengcheng meteor radar
transmitter.
The Changfeng remote receiver system consists of five receiver antennas using a 'T'
shaped arrangement (Jones et al., 1998), a digital transceiver identical to the
Mengcheng meteor radar. To permit accurate range and Doppler estimates at the remote
site, the system timing, frequency, and clocks at both sites are synchronized with GPS-
disciplined oscillators (GPSDOs). The techniques used to estimate various data
products from the received meteor echoes, including radial velocity, meteor position,
and decay time, follow those outlined in Holdsworth et al. (2004a). Both radars belong
to and are operated by the University and Science and Technology of China (USTC).
The dataset considered spans 6 days from 15 October to 20 October 2021.
**3 Observations**
Figures 2a and 2b show the histograms of meteor height distribution observed by the
MCR and CFR, which are well approximated by a fitted Gaussian curve (as shown by
red dashed curves). The peak heights of the meteor height distribution observed by
MCR and CFR are approximately 90 and 91 km, respectively. The peak height of the
CFR appears to be 1 km higher than that of the MCR because the equivalent frequency
or effective Bragg wavelengths for the forward scatter of the CFR would be lower than
those for the backscatter detected by the MCR. The results of the equivalent frequency
will be presented later. Figure 2c shows that the hourly meteor number observed by the
MCR is larger than that of the CFR. The meteor number of the forward scatter observed
by the CFR is approximately 71% of the detections using backscatter by the MCR
monostatic system. These results are similar to the results from two bistatic meteor





radar systems reported by Stober et al. (2015) and Spargo et al. (2019). In addition, the
meteor count rates observed by the MCR and CFR both show a clear diurnal variation,
with a high-count rate in the local morning (i.e., 2000-0004 UT) and a low count rate
during local night (i.e., 8000-1600 UT).
Figure 3 shows the projection of meteor detections observed by the MCR and CFR on
a plan view map. Figure 3a shows a reasonable overlap of meteor detections over the
two receiving sites. In Figures 3b and 3d, the backscattered echoes are observed in a
roughly circular region with an approximately 300 km radius and are mainly distributed
50-120 km from the MCR receiver. In Figures 3c and 3e, the forward scatter meteor
echoes observed by the CFR are more widely and evenly distributed than those
observed by single-station radar, and the meteors are mainly distributed within a
circular region of radius of about 100 km to the CFR.

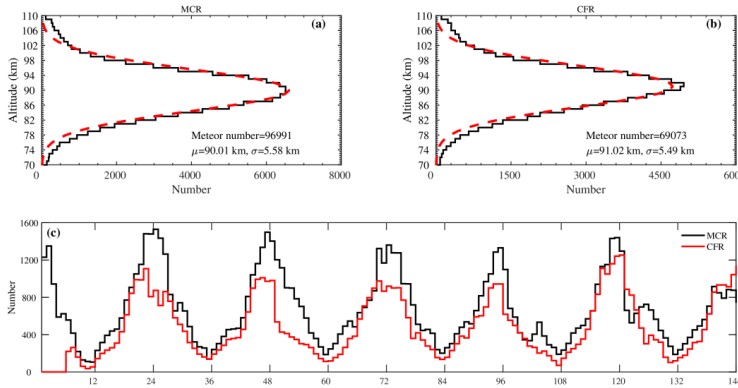


**Figure 2.** The height distribution of meteor detections in 1 km bins during October 15-
20, 2021, observation by (a) Mengcheng and (b) Changfeng receivers. The fitted
Gaussian curves used for the estimation of peak height (μ) and standard deviation (σ)
of meteor height distribution. (c) Hourly meteor numbers observed by the Mengcheng
(black line) and Changfeng (red line) radars.

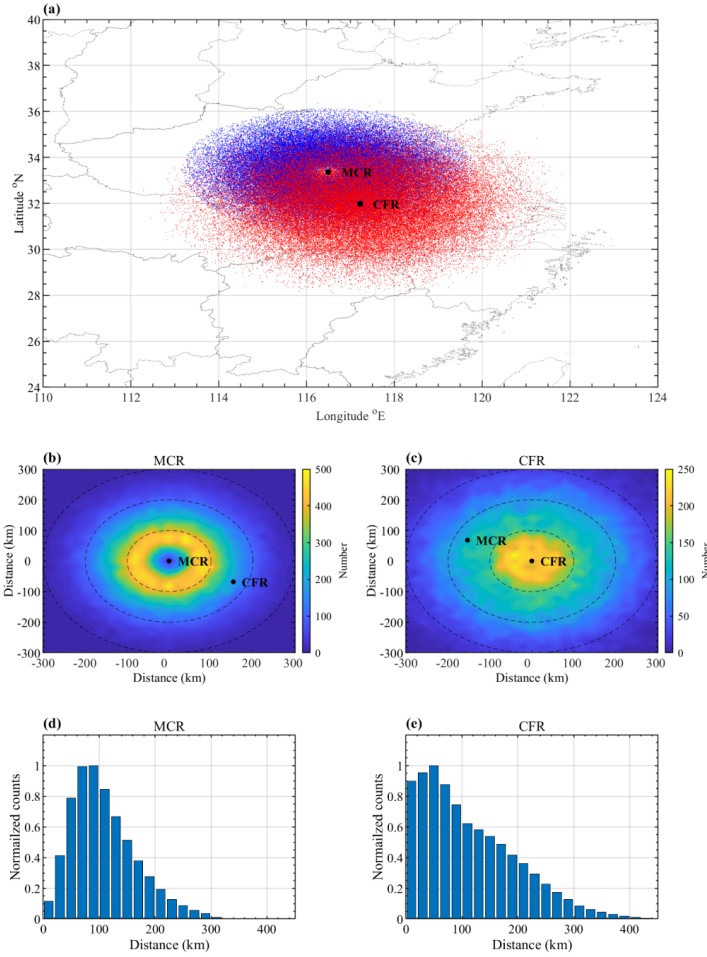


**Figure 3.** (a) Two-dimensional projection of meteor detections observed by Mengcheng (blue dots) and Changfeng (ret dots) receivers. Horizontal distribution of meteors for the (b) Mengcheng and (c) Changfeng receivers. Histograms of meteor number ratio versus distance observed by the (d) Mengcheng and (e) Changfeng receivers. The distance represents the horizontal distance from the projection of meteor echoes to receivers.

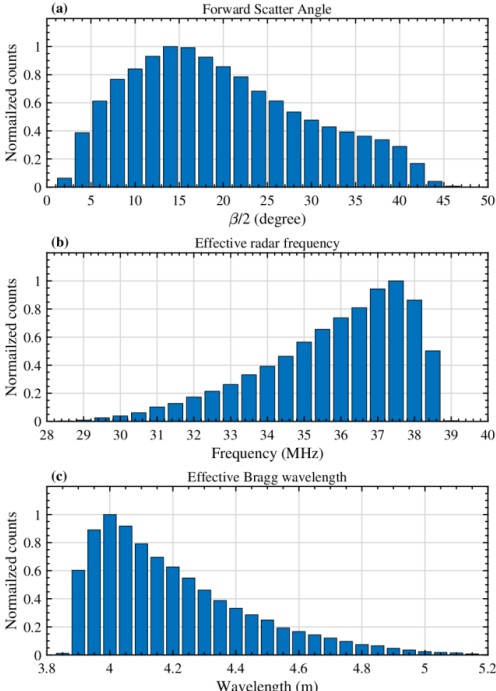

**Figure 4.** Histograms of detections as a function of (a) forward scatter angle, (b) equivalent frequencies and (c) effective Bragg wavelength for the Changfeng remote receiver.

Returning now to the geometrical relationship shown in Figure 1, $\vec{t}$ and $\vec{b}$ represent the vectors from the MCR transmitter to meteor and the meteor to the CFR receiver, respectively. $\vec{d}$ represents the straight line from the MCR transmitter to the CFR receiver. The angle of arrival (AOA, i.e., zenith and azimuth) for the CFR remote receiver can be calculated by using the phase differences between the interferometric antennas. However, the angle ($\alpha$) between the scatter wave ($\vec{t}$) and $\vec{d}$ (the straight line from the MCR to CFR) has a small difference from the elevation angle of the scatter wave because of the Earth's curvature. This small difference can be calculated using the geometry of Earth's curvature and radius and the location of the transmitter and





receiver. Then, the path length of incident and scattered waves can be calculated using
Equation (1) (Doviak and Zrnic, 2006).

$$|\vec{t}| = \frac{R_i^2 - |\vec{d}|^2}{2 \cdot (R_i - |\vec{d}| \cos(\alpha))}, \qquad (1)$$

where $R_i$ is the range of total wave path $R_i = |\vec{t}| + |\vec{b}|$ from the MCR transmitter to
the meteor trail and to the CFR receiver. $R_i$ is given by $R_i = R + iR_{amb}$, $R_{amb} =$
$c/(2 \cdot PRF)$ is the maximum unambiguous range, $c$ is the speed of light, PRF for the
Mengcheng meteor radar is 430 Hz, and the typically unambiguous number $i$=0, 1, 2,….
Therefore, for the Mengcheng meteor radar, the maximum echo range is $R_{amb} = 349$
km, and the unambiguous number is estimated using $i$=0 or 1 (e.g., Holdsworth et al.,

209  2004).

$$\beta = \cos^{-1}\left(\frac{\vec{t} \cdot \vec{b}}{|\vec{t}||\vec{b}|}\right), \qquad (2)$$

The forward scatter angle can be estimated by using equation (1) (e.g., Stober et al.,
2018, Spargo et al., 2019). As shown in Figure 4a, the forward scatter angle ($\beta/2$)
values vary between 0° and 50°. The lower value of the forward scatter angle is close
to 0°; in this case, the scattering geometry is similar to that of the backscatter model.
The larger value of 50° corresponding to the meteor trail is between the MCR
transmitter and CFR receiver at 70 km altitude. The meteor trails are concentrated at
approximately 15°, which means that the meteor trails are mainly distributed over the
CFR receiver.
Figure 4b shows the distribution of equivalent frequencies corresponding to the meteor
trail observed by the Changfeng receiver. The equivalent frequencies show a peak at
approximately 37.5 MHz, which is 1.4 MHz lower than the Mengcheng transmitted
frequency (38.9 MHz). The lowest equivalent frequencies are approximately 28.5 MHz,
so the frequency bandwidth is approximately 10.4 MHz. This result explains why the
peak of the meteor height distribution observed by the CFR receiver is approximately
1 km higher than that of the backscatter meteor trails observed by the MCR receiver
(Ceplecha et al., 1998; Yi et al., 2018). Stober and Chau (2015) transmitted two
frequencies at 32.55 MHz and 53.5 MHz and observed by a 118 km remote receiver

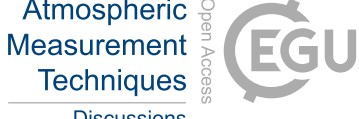



with two peaks (bandwidth) frequencies at approximately 31 (5.5) MHz and 49 (10.5)
MHz, respectively. This finding is consistent with the suggestion that a higher
transmitter frequency gives a wider equivalent frequency bandwidth (e.g., Stober et al.,

231   2015).

The projected velocity of the meteor trail observed by the remote station is considered
along the Bragg wave vector (*de Elía and Zawadzki*, 2001). Therefore, for the forward
scatter geometry, the direction of the radial drift velocity of the meteor trail represents
the Bragg wave vector, i.e., $\vec{e}$ shown in Figure 1 (e.g., Stober et al., 2015; Spargo et al,
2019). The Bragg wave vector $\vec{e}$ can be obtained as $\vec{e} = \frac{\vec{t} - \vec{b}}{|\vec{t} - \vec{b}|}$. In the case of the
backward scatter geometry based on the monostatic meteor radar, the direction of radial
drift velocity is perpendicular to the meteor trail. The Bragg wavelengths of backscatter
are $\frac{\lambda}{2} = 3.86$ m. For the forward scatter geometry, the Bragg wavelengths are given by

$$\lambda_B = \frac{\lambda}{2\cos(\beta/2)},\tag{3}$$

In Figure 4c, the Bragg wavelength distribution shows a peak at approximately 4 km
with a bandwidth of approximately 1.4 m. The radial drift velocity projected along the
Bragg wavelengths measured by the remoter receiver can be expressed as

$$v_B = f_d \lambda_B,\tag{4}$$

where $f_d$ is the Doppler frequency shift. In the case of backward scatter geometry, the
radial velocity is perpendicular to the meteor trial and is $v_B = f_d \lambda/2$.
Figure 5 shows the angle of arrival (zenith and azimuth) of meteor echoes and the Bragg
vector observed by the Mengcheng and Changfeng receivers, respectively. The angles
of arrivals observed by the MCR and CFR receivers are basically a similar distribution,
the zenith angles are mainly distributed from 45°-70°, and the azimuth angles are
relatively evenly distributed, with a slightly greater number in the area north (i.e., 350°-
20°) of the receivers. The bistatic meteor radar distribution provides a large increase in
scattering detections per unit time along with observations of the same volume from
different viewing angles.

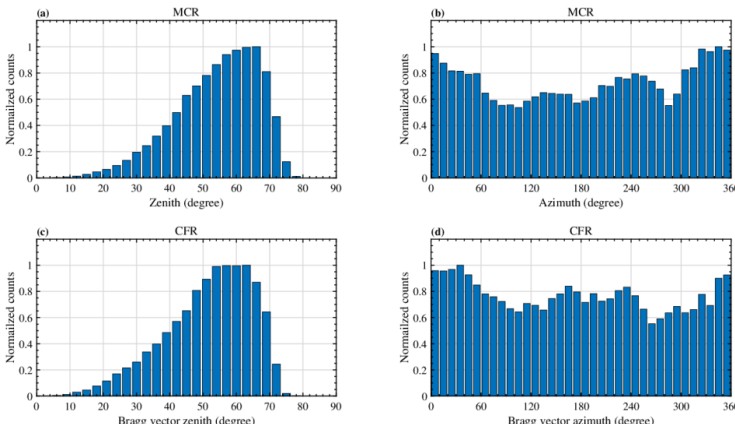

**Figure 5.** Histograms of (a) zenith and (b) azimuth of meteor echoes observed by the Mengcheng (upper) and the (c) zenith and (d) azimuth of the Bragg vector observed by the Changfeng (lower) receivers.

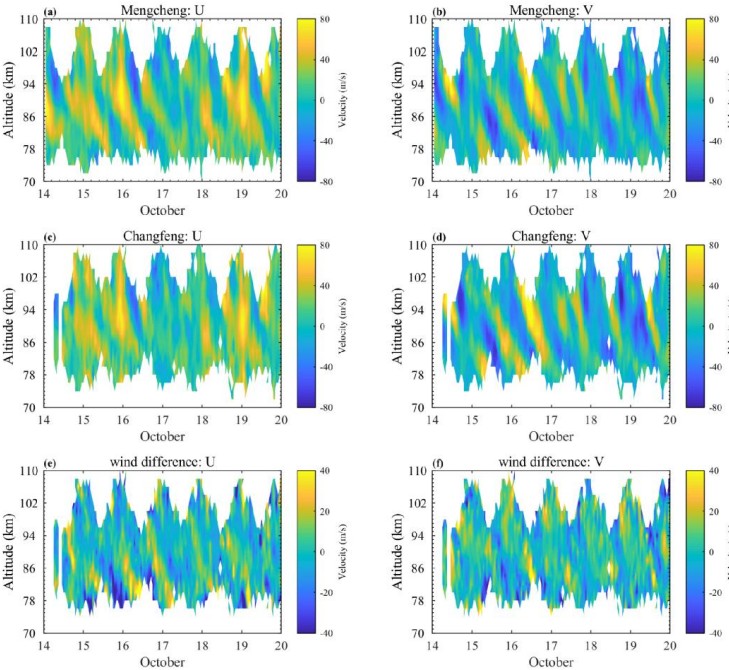

**Figure 6.** Contour plots of (right) zonal and (left) meridional hourly mean winds observed by the (upper) Mengcheng and (middle) Changfeng receivers and (lower) the



wind differences between the Mengcheng and Changfeng measurements.
Given the angle of arrival and radial velocity, the averaged horizontal wind can be
estimated by both monostatic and remote receivers. Figure 6 shows the comparison of
the hourly averaged horizontal (zonal and meridional) winds observed by the MCR and
CFR. The zonal and meridional mean winds all show dominant diurnal (24 h) variations
with a clear upward propagating phase. In Figures 6e and 6f, the wind differences
between the two measurements still show a weak tidal structure, which may be because
the tidal waves over the two receiver sites are different and the two receivers measuring
different viewing areas of the atmosphere make slightly different geophysical waves,
especially tidal waves. A comparison study to discuss tidal differences observed by the
bistatic meteor radar system described in this study and the two collocated meteor
radars at Kunming station (Zeng et al., 2022) is working on. This provides an
exploration of the viewing geometry and geophysical volume, and the diurnal variation
of the meteor count effecting on tide estimation. However, discussion of these
differences is beyond the scope of the present study and will be discussed in the future
paper.
**4. Two-dimensional horizontal wind observed by the bistatic meteor radar**
The averaged wind components are generally calculated by applying a so-called all-sky
method under the assumption of a homogeneous wind field (e.g., Hocking et al., 2001,
Holdsworth et al., 2004). The multistatic geometry allows the investigation of nearly
the same phenomenon from different angles and volumes and thus makes it possible to
reveal the inhomogeneities of the wind fields. Browning and Wexler (1968) introduced
the velocity azimuth display (VAD) method for situations in which the wind field is not
horizontally uniform, applying a linearity hypothesis to acquire the horizontal winds
with their derivatives. Waldteufel and Corbin (1979) proposed the Volume Velocity
Processing (VVP) method, making full use of the meteor echoes within the observation
volume to obtain the linear wind field. The difference between these two approaches
mainly lies in the idea of solving equations. Stober et al. (2013) compared both methods
in terms of gravity wave detection and found no distinct difference between them. In





this study, we apply the VVP method to retrieve the horizontal winds.
According to the VVP method, the wind components of the scatter motion V = (u, v, w)
can be described linearly by

$$u(x, y, z) = u_0 + \frac{\partial u}{\partial x}(x - x_0) + \frac{\partial u}{\partial y}(y - y_0) + \frac{\partial u}{\partial z}(z - z_0),$$
$$v(x, y, z) = v_0 + \frac{\partial v}{\partial x}(x - x_0) + \frac{\partial v}{\partial y}(y - y_0) + \frac{\partial v}{\partial z}(z - z_0), \tag{6}$$
$$w(x, y, z) = w_0 + \frac{\partial w}{\partial x}(x - x_0) + \frac{\partial w}{\partial y}(y - y_0) + \frac{\partial w}{\partial z}(z - z_0),$$

where (x, y, z) are the coordinates in the Cartesian reference frame and $(u_0, v_0, w_0)$ is
the mean wind at a fixed point $(x_0, y_0, z_0)$. In stratiform situations, it is appropriate to
ignore $\frac{\partial w}{\partial x}$ and $\frac{\partial w}{\partial y}$ with respect to $\frac{\partial u}{\partial z}$ and $\frac{\partial v}{\partial z}$ (Waldteufel and Corbin, 1979). Here
we only focus on the horizontal components; thus, we assume $w_0 = 0$ and $\frac{\partial w}{\partial z} = 0$
for the simplicity of the equations and obtain

$$u(x, y, z) = u_0 + \frac{\partial u}{\partial x}(x - x_0) + \frac{\partial u}{\partial y}(y - y_0) + \frac{\partial u}{\partial z}(z - z_0),$$
$$v(x, y, z) = v_0 + \frac{\partial v}{\partial x}(x - x_0) + \frac{\partial v}{\partial y}(y - y_0) + \frac{\partial v}{\partial z}(z - z_0). \tag{7}$$

The radial velocity can be expressed as

$$V_r = u \sin \phi \sin \theta + v \cos \phi \sin \theta, \tag{8}$$

where $\theta$ and $\phi$ are the zenith and azimuth angles, respectively. Using the least square
method, the mean winds and the inhomogeneities of the winds (such as the horizontal
divergence $(\frac{\partial u}{\partial x} + \frac{\partial v}{\partial y})$, relative horizontal vorticity $(\frac{\partial v}{\partial x} - \frac{\partial u}{\partial y})$, stretching $((\frac{\partial u}{\partial x} - \frac{\partial v}{\partial y})$ and
shearing $(\frac{\partial u}{\partial y} + \frac{\partial v}{\partial x})$ can be easily achieved due to the large number of meteor echoes
detected in the selected volume.
We set the MCR and CFR as the origins of the two local ENU coordinates, and for the
convenience of the calculation, we assume the midpoint of the two stations as the fixed
point, that is $(x_0, y_0) = (x_{mid}, y_{mid})$, and $z_0$ is the reference altitude, normally
ranging from 70 km to 110 km. The meteor locations $(x_m, y_m, z_m)$ in both two local
ENU coordinates are calculated using the detected range and arrival angle, and then
considering the curvature of the Earth, we conduct transformations as described by
Stober et al.(2018). First, we transform the meteor location $(x_m, y_m, z_m)$ into the ECEF
coordinates $(X_M, Y_M, Z_M)$. Then convert the ECEF coordinates $(X_M, Y_M, Z_M)$ into
geodetic coordinates $(\phi_m, \lambda_m, h_m)$. Finally, the local ENU coordinates of meteor
echoes to the midpoint can be calculated as $(x_m', y_m', z_m')$. We conduct 2-D wind fitting
by shifting a [3 km, 1 h] window by a [1 km, 0.25 h] step. The windows are centered at
the interested height and time, containing no less than 10 meteors for the accuracy of
the retrieval. Then, using the meteor information relative to the two stations and
applying the least squares method, the 8 unknowns in Equation (7) can be retrieved,
and we can select the area of interest to estimate the local wind fields. Note that the 8
unknowns are corresponding to the whole area, and the local winds are calculated using
Equation (6) for a given point (x, y, z).

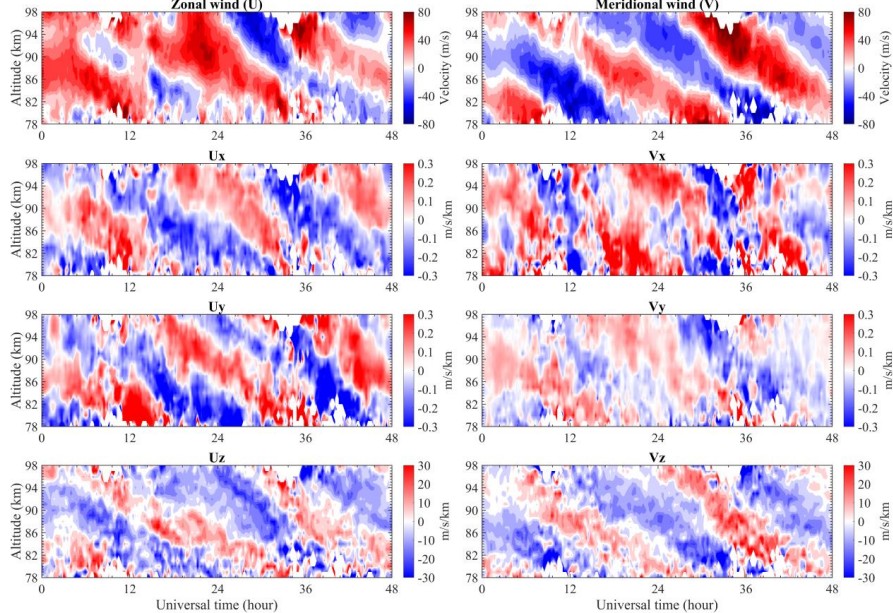


**Figure 7**. Mean horizontal winds and the gradient terms of the MLT wind fields on
October 16 and 17, 2021. The left side represents the zonal component, and the right
side represents the meridional component.
Figure 7 shows the mean winds $(u_0, v_0)$ and the gradient terms $(u_x, u_y, u_z, v_x, v_y,$
$v_z)$ of the horizontal wind fields on October 16 and 17, 2021, retrieved using the MCR-
CFR composite data sets. These 8 parameters are fitted using Equations (7) and (8) with



the meteor information, which is the location of the meteors relative to the MCR and
the corresponding radial velocity vector of the CFR output. The mean winds present a
diurnal tidal structure, and their horizontal and vertical gradient terms also show distinct
diurnal signatures, though $v_x$ seems to show diurnal/semidiurnal features
below/above 84 km. The magnitude of the gradient terms is nearly the same, and the
values of $v_y$ are relatively smaller. Chau et al. (2017) calculated the wind parameters
in the polar region and exhibited similar semi-diurnal results.
In order to verify the reliability of our results, we compared the traditional all-sky
results and the VVP results by calculating the correlation coefficients and the regional
winds. The correlation coefficients are shown in Figure 8. The upper row shows the
VVP mean winds versus the all-sky mean winds, and the bottom row shows the VVP
vertical gradients versus the calculated all-sky vertical gradients using the mean winds
from the adjacent time-height bins. The correlation between the mean winds retrieved
by these two methods is higher than 0.9, illustrating high consistencies, and the
correlations between the computed $u_z$, $v_z$ and the calculated derivative of horizontal
winds in vertical direction are also appreciable, both verifying the reliability of the
gradient results using VVP method. Besides, by careful observation of the normalized
density distribution, we find the meridional terms ($u_z$, $v_z$) are more symmetrically
distributed.



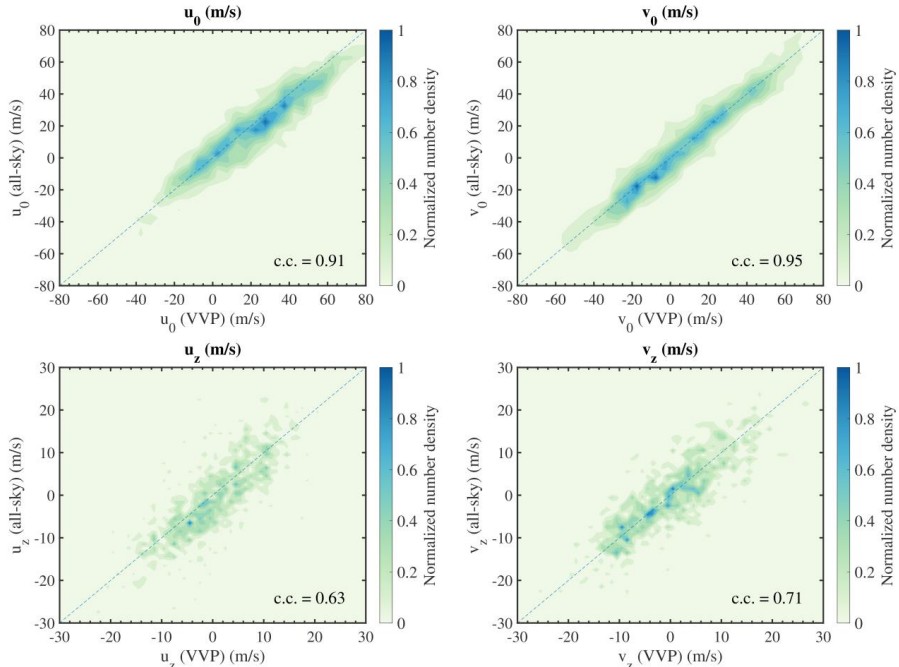

**Figure 8.** The correlation coefficients between the VVP method (x-axis) and the all-sky method (y-axis). The upper row shows the mean zonal wind and mean meridional wind correlations, and the bottom row shows the correlation of vertical gradients of zonal and meridional wind components. The dashed blue line represents x=y. The blue-green blocks are the normalized number density. The values of correlation coefficients are labeled respectively.

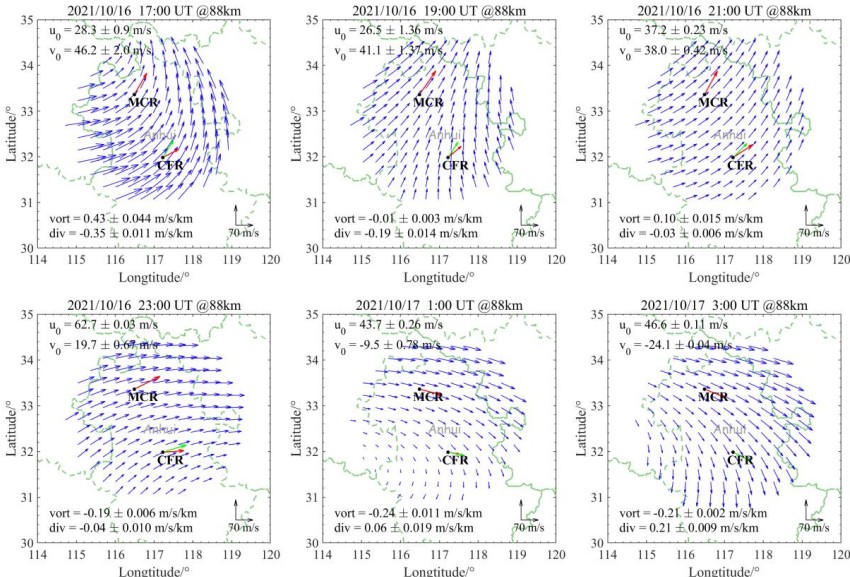

**Figure 9.** Hourly wind fields at 88 km from 1700 UTC on 16 October 2021 to 0300
UTC on 17 October 2021. The green dashed line shows the provincial boundaries of
Anhui Province. The blue arrows represent the wind vectors. The red and green arrows
represent the horizontal mean winds calculated using the all-sky method and VVP
method, respectively. The value of the mean winds, vorticity, divergence, and their
uncertainties are also labeled.

Figure 9 shows the temporal evolution of the wind field at 88 km from 1700 UTC on
16 October 2021 to 0300 UTC on 17 October 2021 for 10 hours, at 2-hour intervals.
The blue arrows represent the wind vector of the grid cell separated by 30×30 km. The
red and green arrows are the winds retrieved by the all-sky method and the VVP method,
respectively. The mean winds rotate clockwise with time, revealing tidal characteristics.
As shown in Figure 9, when the wind field is nearly homogeneous, such as 2100 UTC
on 16 October 2021, the derived wind fields are almost identical to the mean winds.
And even when the wind field shows an obvious vortex structure, the derived regional
wind fields and the averaged winds are well matched.



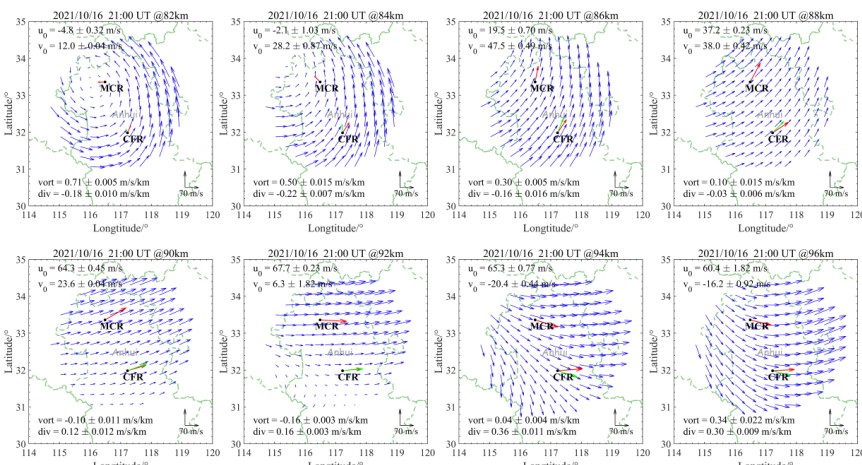

374

**Figure 10.** Hourly wind fields for different heights (82, 84, 86, 88, 90 km) at the same time (1900 UTC on 17 October 2021). The meaning of the symbols is the same as in Figure 9. The value of the mean winds, vorticity, divergence, and their uncertainties are also labeled.

Figure 10 presents the height evolution of the wind field at 2100 UTC on 16 October 2021. The winds show distinct vortex structures at 82 and 84 km, and become more homogeneous at higher altitudes. Comparing the all-sky mean winds (red arrows) with the VVP regional winds (blue arrows) carefully, we can find the wind magnitudes and directions are in excellent agreement, even when there are strong vortex structures. Looking at the regional winds in order of height, the characteristics of the wind changes is similar to the temporal evolution of the wind fields, which is also a change of phase. The phase variation characteristics of the wind fields in height corroborates the diurnal structure in $u_z$ and $v_z$.

388



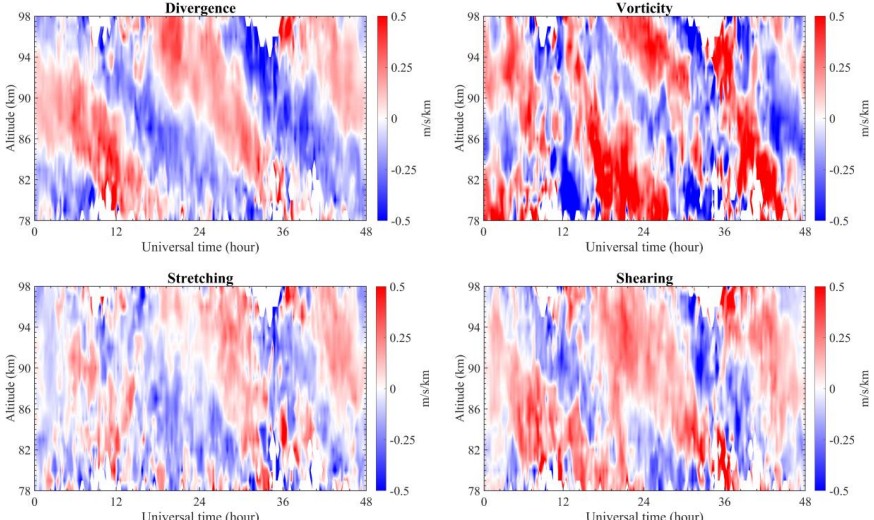

389

**Figure 11.** The horizontal divergence (upper left), relative vorticity (upper right), stretching deformation (lower left) and shearing deformation (lower right) calculated from the horizontal gradients of the horizontal wind.

In order to appreciate better the horizontal wind parameters, we calculate the horizontal divergence, relative vorticity, stretching deformation and shearing deformation using the horizontal gradients of the horizontal wind. As shown in Figure 11, the horizontal divergence shows dominant diurnal variations with a clear upward propagating phase. The diurnal variation structure is similar to the zonal wind shown in Figure 6. Qualitatively, the zonal eastward/westward winds may correspond to the positive/negative horizontal divergence values. The relative vorticity shows more complicated vertical structures compared to the horizontal divergence. The relative vorticity mainly shows the semidiurnal/diurnal variations above/below 84 km. The shearing deformation is associated with the reversal of the winds, and also shows diurnal features. The characteristics of the stretching deformation are similar to that of the shearing deformation. However, the inherent relationship between the horizontal wind parameters and dynamics in the MLT region is still not clear and needs further exploration.

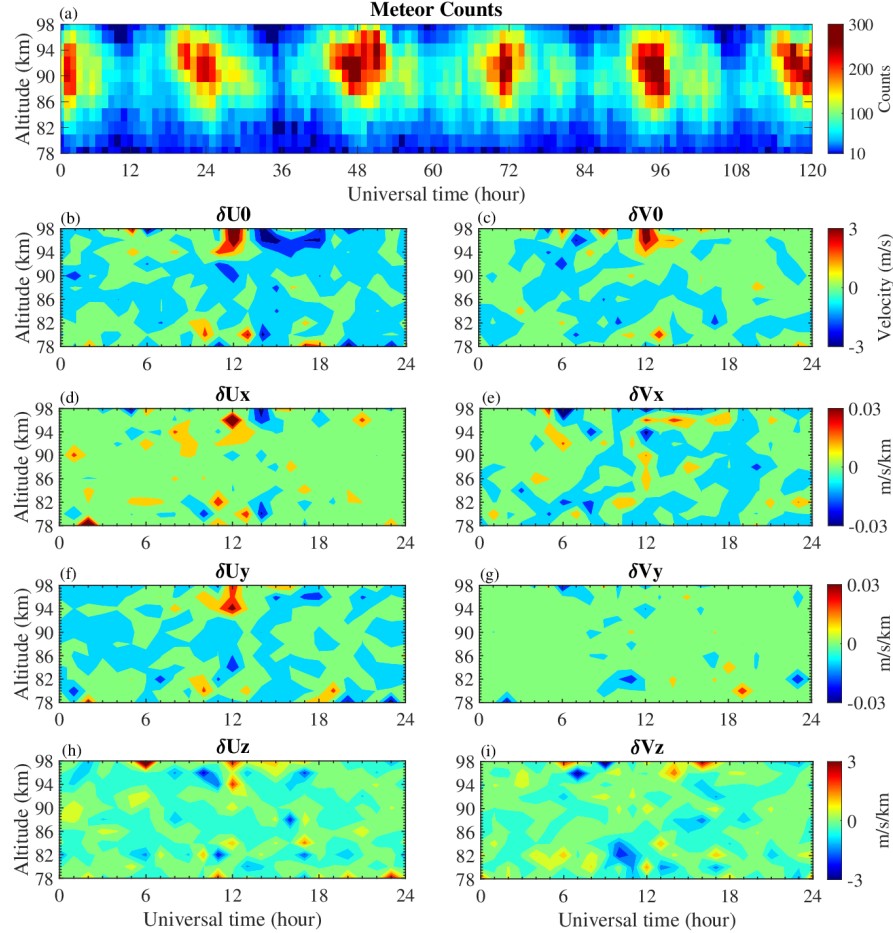

**Figure 12.** (a) The valid meteor counts on October 16 to October 20, 2021. (b-i) The errors of horizontal winds and gradient terms corresponding to a composite day (October 16 to October 20, 2021).

We stated that at least 10 meteors would be needed for estimations, and actually, there are more meteor echoes involved in the calculation, which is shown in Figure 12 (a). The valid meteor count is the number of meteors actually used in the last calculation process after several iterations. It is clear that the valid meteor counts are larger than 10 in [78km, 98km], and the errors due to the lack of meteor detections may be deduced. As shown in Figure 12, we estimate the errors in winds and gradient terms using the radial velocity error estimation obtained by the radar system (Holdsworth et al., 2004a). The composite error estimations utilizing the data from October 16 to October 20, 2021



are shown in Figures 12 (b-i). It is clear that the errors of horizontal winds and gradient
terms are smaller than 1m/s and 0.1 m/s/km, respectively, when the meteor detections
are sufficient, such as the results ranging from 82 to 94 km during the local morning
(2000-0004 UT). The large errors basically occur above 94 km during the local night
(near 1200 UT), which is mainly caused by a low number of meteors.
Our results verify the ability of the VVP method to estimate the wind parameters. And
based on these parameters, multistatic meteor radars are capable of deducing the
inhomogeneities and kinetic characteristics of the wind fields, which are similar to those
of Stober and Chau (2015). The increased meteor detections can reduce the error of the
estimated terms and guarantee the reliability of the results. Subsequent work focusing
on these specific dynamics will be reported in the future.
**5. Discussion and Summary**
In this study, we have presented the preliminary results from the Mengcheng and
Changfeng bistatic meteor radar systems. The main objectives were accomplished
successfully by the new bistatic meteor radar system and are summarized as follows:
1. The bistatic meteor radar system consists of a conventional meteor radar located at
Mengcheng and a remote receiver located at Changfeng. The remote receiver observes
the forward scatter meteor echoes transmitted from the Mengcheng transmitter.
Compared to the monostatic meteor radar operation, we detect ~70% more forward
scatter meteor detections by using the bistatic radar system. In addition, the forward
scatter meteor echoes provide different viewing angles of the radial velocity and a larger
viewing area of the atmosphere compared to the monostatic backscatter meteor radar.
2. Based on a distance of 167.3 km between the radar transmitter and remote receiver,
those quantities depending on the geometry of the forward scatter arrangement, such as
the forward scatter angle, equivalent frequencies and Bragg wavelength, were estimated.
The forward scatter meteor echoes are normally ~1 km higher than the backscatter
meteor echoes because the equivalent frequencies (the effective Bragg wavelengths) of
forward scattering are lower (larger) than those of backscatter meteor echoes (Ceplecha
et al., 1998). The bistatic meteor radar system generally provides a more than 400



km×400 km horizontal viewing area.
3. Taking advantage of the increased meteor number and different viewing angles
observed by the bistatic meteor radar system, we can relax the assumption of a
homogeneous horizontal wind and estimate the two-dimensional horizontal wind more
accurately using the volume velocity processing method. The improved wind
estimation provides the mean winds and the inhomogeneities of the winds (such as the
horizontal divergence, relative horizontal vorticity, stretching and shearing
deformation).

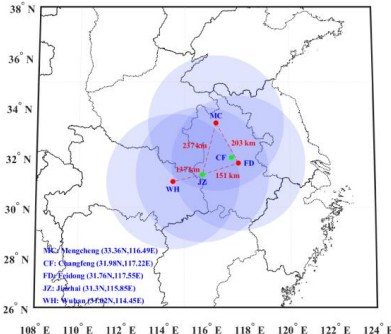


**Figure 13.** Design schematic view of the multistatic meteor radar network. The red dots
represent the monostatic meteor radars located in Mengcheng, Feidong and Wuhan. The
green dots represent the remote receivers at Changfeng and Jinzhai. The distance
between each site is marked. The blue-shaded areas label a circle of 300 km in diameter
around each center of radial velocity measurements.
The preliminary results of the MLT region observed by our bistatic meteor radar system
are encouraging and lay a foundation for an extension of the new multistatic meteor
radar network. Figure 13 shows the design schematic view of the new multistatic
meteor radar system in Central-Eastern China. At present, we are installing a new
monostatic meteor radar at Feidong (31.76 °N, 117.55°E). The distance between the
Feidong and Mengcheng meteor radars is approximately 203 km, which will enable
transmitting and receiving of Mengcheng and Feidong meteor radars from each other.





Then, the Changfeng remote receiver will be moved to Jinzhai, and the Jinzhai site will
be able to receive the forward scatter meteor echoes from the Mengcheng, Feidong and
Wuhan (e.g, Zhou et al., 2022) meteor radars, which all operate at 38.9 MHz. The new
multistatic meteor radar system will achieve three backscatter (monostatic) links and
five forward scatter (bistatic) links, which would provide us with 6 times more meteor
detections than a conventional (monostatic) meteor radar.
The new multistatic meteor radar network will provide a better determination of the
horizontal and vertical gradients of the horizontal winds by increasing the meteor
number and extending the atmospheric viewing area, which allows us to investigate
gravity waves with horizontal scales smaller than hundreds of kilometers (e.g., Stober
et al., 2018, 2021b, 2022; Conte et al., 2020) and estimate a higher temporal resolution
of the standard horizontal mean wind (e.g., Vierinen et al., 2019; Vargas et al., 2021;
Zhong et al., 2021). Moreover, the multistatic meteor radar network can estimate not
only the mean horizontal and vertical winds but additional quantities, such as the
horizontal divergence, relative vorticity, stretching, and shearing deformation of the
wind field (e.g., Stober et al., 2015; Chau et al., 2017, 2020; Volz et al., 2021). In
addition, multistatic meteor radar data can also be used to investigate smaller-scale
MLT perturbations by estimating the second-order statistics of radial velocities, for
example, the gravity wave momentum fluxes in the MLT region (e.g., Spargo et al.,
2019). Furthermore, the decay time or ambipolar diffusion coefficient of meteor trails
measured by multistatic meteor radar can be used to estimate the mesospheric neutral
temperature, pressure and density (e.g., Hocking et al., 1997; 1999; Younger et al., 2015;
Yi et al., 2019), as well as the mesospheric ozone density (Sukara, 2013). The velocity
and spatial position information of meteors can be used for meteor orbit and meteor
shower detection (e.g., Holdsworth et al., 2007; Younger et al., 2015). We are confident
that the multistatic meteor radar network system is a powerful technique for achieving
comprehensive observation of the MLT region and will provide an opportunity to
understand MLT dynamics.
***Author contributions.*** WY and XX designed the study. WY and JZ carried out the data



analysis and wrote the paper. XX supervised the work and provided valuable comments. IMR revised the paper. All of the authors discussed the results and commented on the paper.

**Acknowledgments:** Wen Yi acknowledge the technical support of our radar systems by Chris Adami and Jinsong Chen. We would like to thank Gunter Stober and Zishun Qiao for useful discussions regarding this work.

***Financial support.***

*This work was supported by* the National Natural Science Foundation of China (grants No. 42125402, 41974174, 42188101, 41831071, 42174183 and 41904135), the B-type Strategic Priority Program of CAS (grant No. XDB41000000), the Project of Stable Support for Youth Team in Basic Research Field, CAS (grant No. YSBR-018), the Fundamental Research Funds for the Central Universities (grant No.YD3420002004), the Joint Open Fund of Mengcheng National Geophysical Observatory (MENGO-202209), the Anhui Provincial Natural Science Foundation (grant no. 2008085MD113), the Open Research Project of Large Research Infrastructures of CAS - "Study on the interaction between low/mid-latitude atmosphere and ionosphere based on the Chinese Meridian Project.

**Data Availability Statement:** The data presented in this study are available on request from the author (Y.W., yiwen@ustc.edu.cn). The data are not publicly available due to institutional restrictions.

**Conflicts of Interest:** The authors declare no conflict of interest.

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
