# Peer review of "Multistatic meteor radar observations of two-dimensional horizontal MLT wind"

_Atmospheric Measurement Techniques, 2022_

## Referee Comment (RC1)

Review of manuscript amt-2022-254, entitled

**Multistatic meteor radar observations of two-dimensional horizontal MLT wind**

By Wen Yi et al.

Submitted to Atmospheric Measurements Techniques (AMT).

This manuscript presents some preliminary results on mesosphere and lower thermosphere (MLT) winds and wind gradients based on measurements obtained from a bi-static meteor radar located in Central China. Overall, the paper is well written and understandable. However, and as I stated in the rapid access review stage, the authors have chosen the wrong journal to submit it.

The manuscript does not contribute with any new information/results on measurement techniques of the atmosphere. Multi-static meteor radars have already been well described in many previous studies (Stober & Chau 2015; Chau et al., 2017; Stober et al., 2018; Vierinen et al., 2019; Spargo et al., 2019; Chau et al., 2021; Poblet et al., 2022; to name only a few). Furthermore, many of these previous studies presented new results on the physics of the MLT, which could be done with the system presented in the manuscript here reviewed, but sadly, it is not done. Only a few days of wind data are presented and briefly discussed. Given that the MLT over Central China has not been studied using multi-static meteor radars, the authors have a unique opportunity to present new and exciting results on MLT dynamics over this part of the world. I encourage them to do so, and resubmit their manuscript to the adequate journal (e.g., Annales Geophysicae or ACP).

- Stober, G., and J. L. Chau (2015), A multistatic and multifrequency novel approach for specular meteor radars to improve wind measurements in the MLT region, Radio Sci., 50, 431–442, doi:10.1002/2014RS005591.
- Chau, J. L., G. Stober, C. M. Hall, M. Tsutsumi, F. I. Laskar, and P. Hoffmann (2017), Polar mesospheric horizontal divergence and relative vorticity measurements using multiple specular meteor radars, Radio Sci., 52, 811–828, doi:10.1002/2016RS006225.
- Stober, G., Chau, J. L., Vierinen, J., Jacobi, C., and Wilhelm, S.: Retrieving horizontally resolved wind fields using multi-static meteor radar observations, Atmos. Meas. Tech., 11, 4891–4907, https://doi.org/10.5194/amt-11-4891-2018, 2018.
- Vierinen, J., Chau, J. L., Charuvil, H., Urco, J. M., Clahsen, M., Avsarkisov, V., et al. (2019). Observing mesospheric turbulence with specular meteor radars: A novel

method for estimating second-order statistics of wind velocity. Earth and Space Science, 6. https://doi.org/10.1029/2019EA000570.

- Spargo, A. J., Reid, I. M., and MacKinnon, A. D.: Multistatic meteor radar observations of gravity-wave–tidal interaction over southern Australia, Atmos. Meas. Tech., 12, 4791–4812, https://doi.org/10.5194/amt-12-4791-2019, 2019.

- Chau, J. L., Urco, J. M., Vierinen, J., Harding, B. J., Clahsen, M., Pfeffer, N., et al. (2021). Multistatic specular meteor radar network in Peru: System description and initial results. Earth and Space Science, 8, e2020EA001293. https://doi.org/10.1029/2020EA001293.

- Poblet, F. L., Chau, J. L., Conte, J. F., Avsarkisov, V., Vierinen, J., & Charuvil Asokan, H. (2022). Horizontal wavenumber spectra of vertical vorticity and horizontal divergence of mesoscale dynamics in the mesosphere and lower thermosphere using multistatic specular meteor radar observations. Earth and Space Science, 9, e2021EA002201. https://doi.org/10.1029/2021EA002201.

---

## Referee Comment (RC4)

**Referee Comments for** amt-2022-254  Submitted on 26 Sep 2022
Multistatic meteor radar observations of two-dimensional horizontal MLT wind
Wen Yi, Jie Zeng, Xianghui Xue, Iain Reid, Wei Zhong, Jianfei Wu, Tingdi Chen, and Xiankang Dou

GENERAL COMMENTS

This paper, "**Multistatic meteor radar observations of two-dimensional horizontal MLT wind**",
introduces a bistatic extension of a traditional all-sky meteor radar system in an effort to estimate two-
dimensional horizontal wind fields by relaxing the usual wind field spatial homogeneity assumption. The
authors provide a good historical context for the work, and highlight recent developments in the field.
The work is generally well organized, but could use additional grammar review. The manuscript meets
basic scientific quality, is free from obvious major deficiencies, and is suitable for peer review.

SPECIFIC COMMENTS

*Angle-of-Arrival*
The Angle-of-Arrival calculations starting on Line 189 and following are well laid out.
1) **Line 192ff**: "AOA can be determined...": Include at least a reference or a brief description of the
process, given that the bi-static case has unique challenges compared to the monostatic case.  This will
help substantiate the statement on Line 344" Through bistatic geometry, the coordinates of the meteor
locations (x, y) can be deduced."
2) **Line 196**: "This small difference can be calculated...": Include an equation and/or a reference to the
procedure.
3) Perhaps discuss any hardware-specific challenges encountered such a calibrating system phase
biases.

*Vertical Components*
**Line 334**: Discuss the validity of setting the vertical component, w0 = 0 especially in light of the addition
information that multistatic meteor radar brings (e.g. multiple viewing angles). Such a simplification is
typical in traditional all-sky meteor radar processing, but perhaps discuss if and why this continues to be
a valid and appropriate approach.

*MLT Dynamics and Other Discussion*
**Line 425ff**: any discussion on previous work, or specific opportunities for exploration over your
geographic area?

**Line 363**: "In order to verify the reliability of our results, we compared the traditional all-sky
results and the VVP results by calculating the correlation coefficients and the regional
winds" If the overall aim of the paper is to relax the homogeneity assumption, presumably to model the
true wind field with better accuracy, is it appropriate to use the correlation between the traditional
method (which is defined as making certain simplifying assumptions) and the VVP method which in
theory should be more accurate? In other words, is the traditional method an appropriate 'ground truth'
metric? In the extreme case, the traditional method's simplifying assumptions are invalid, in which case
validating a new method based on high correlation with the traditional method is inappropriate.

**TECHNICAL CORRECTIONS**

**Line 346**: centred -> centered

**Line 374** might be out of place, the preceding paragraph talks about the correlation between mean winds calculated by the VVP and traditional all-sky method, while the concluding sentence mentions a semidiurnal tide in the polar region.

**Line 420**: This sentence does not flow, suggest a grammar revision: "Qualitatively, the zonal eastward/westward winds are like to corresponding to the positive/negative the horizontal divergence values."

**Line 422**: "shows more complicate" -> "shows more complicated"

**Line 425**: "needs more explored" -> Does not flow.

**Line 450**: "increase in the meteor number" -> Does not flow.

---

## Author Comment (AC1)

We thank the reviewer for the useful suggestions to improve the paper. These comments are all valuable and very helpful for revising and improving our manuscript, as well as the important guiding significance to our research. These changes in the revised manuscript have been marked in the track changes version manuscript, as well as the point-to-point responses have been listed as follows:

**Response to reviewer #4**

**Comments:**

This manuscript presents some preliminary results on mesosphere and lower thermosphere (MLT) winds and wind gradients based on measurements obtained from a bi-static meteor radar located in Central China. Overall, the paper is well written and understandable. However, and as I stated in the rapid access review stage, the authors have chosen the wrong journal to submit it. The manuscript does not contribute with any new information/results on measurement techniques of the atmosphere. Multi-static meteor radars have already been well described in many previous studies (Stober & Chau 2015; Chau et al., 2017; Stober et al., 2018; Vierinen et al., 2019; Spargo et al., 2019; Chau et al., 2021; Poblet et al., 2022; to name only a few). Furthermore, many of these previous studies presented new results on the physics of the MLT, which could be done with the system presented in the manuscript here reviewed, but sadly, it is not done. Only a few days of wind data are presented and briefly discussed. Given that the MLT over Central China has not been studied using multi-static meteor radars, the authors have a unique opportunity to present new and exciting results on MLT dynamics over this part of the world. I encourage them to do so, and resubmit their manuscript to the adequate journal (e.g., Annales Geophysicae or ACP).

Stober, G., and J. L. Chau (2015), A multistatic and multifrequency novel approach for specular meteor radars to improve wind measurements in the MLT region, Radio Sci., 50, 431–442, doi:10.1002/2014RS005591.

Chau, J. L., G. Stober, C. M. Hall, M. Tsutsumi, F. I. Laskar, and P. Hoffmann (2017), Polar mesospheric horizontal divergence and relative vorticity measurements using

multiple specular meteor radars, Radio Sci., 52, 811–828, doi:10.1002/2016RS006225.

Stober, G., Chau, J. L., Vierinen, J., Jacobi, C., and Wilhelm, S.: Retrieving horizontally resolved wind fields using multi-static meteor radar observations, Atmos. Meas. Tech., 11, 4891–4907, https://doi.org/10.5194/amt-11-4891-2018, 2018.

Vierinen, J., Chau, J. L., Charuvil, H., Urco, J. M., Clahsen, M., Avsarkisov, V., et al. (2019). Observing mesospheric turbulence with specular meteor radars: A novel method for estimating second-order statistics of wind velocity. Earth and Space Science, 6. https://doi.org/10.1029/2019EA000570.

Spargo, A. J., Reid, I. M., and MacKinnon, A. D.: Multistatic meteor radar observations of gravity-wave–tidal interaction over southern Australia, Atmos. Meas. Tech., 12, 4791–4812, https://doi.org/10.5194/amt-12-4791-2019, 2019.

Chau, J. L., Urco, J. M., Vierinen, J., Harding, B. J., Clahsen, M., Pfeffer, N., et al. (2021). Multistatic specular meteor radar network in Peru: System description and initial results. Earth and Space Science, 8, e2020EA001293. https://doi.org/10.1029/2020EA001293.

Poblet, F. L., Chau, J. L., Conte, J. F., Avsarkisov, V., Vierinen, J., & Charuvil Asokan, H. (2022). Horizontal wavenumber spectra of vertical vorticity and horizontal divergence of mesoscale dynamics in the mesosphere and lower thermosphere using multistatic specular meteor radar observations. Earth and Space Science, 9, e2021EA002201. https://doi.org/10.1029/2021EA002201.

**Response:** Thank you for your great suggestion. This will encourage us to improve our manuscript, as well as the important guiding significance to our research. There are indeed many previous studies related to new reversal methods/algorithms of multistatic meteor radars. Based on these studies, we are able to implement and make use of the first multistatic meteor radar system in China. In this study, we have presented preliminary results from the Mengcheng and Changfeng bistatic meteor radar systems. As the Reviewer suggested, we tried to improve our analysis based on the bistatic meteor radar system. Therefore, we added a new section to investigate the seasonal variations in MLT dynamics at lower-midlatitudes (near $30\,°N$) in the Northern

Hemisphere. Since this is the first observation in the mid-latitudes, we still lack other observations in this region to verify the validity of our observations.

Moreover, the preliminary results of the MLT region observed by the bistatic meteor radar system are encouraging and lay a foundation for an extension of the new multistatic meteor radar network in Central-Eastern China. The new multistatic meteor radar network will provide a better determination of gravity waves with horizontal scales smaller than hundreds of kilometers (e.g., Stober et al., 2018, 2021b, 2022; Conte et al., 2020; Poblet et al., 2022) and estimate a higher temporal resolution of the standard horizontal mean wind (e.g., Vierinen et al., 2019; Vargas et al., 2021; Zhong et al., 2021).

*Here is what we have added in the revised manuscript.*

**5. Seasonal variation dynamics parameters in the MLT region**

This section uses the methodology described in the previous section to estimate the seasonal variation in MLT dynamics parameters, including horizontal wind deformations and gravity wave (GW) momentum fluxes, in the MLT region in the middle latitudes of the Northern Hemisphere from February 2022 to February 2023. As shown in Figure 13, we present the seasonal variations in monthly mean horizontal winds, horizontal divergence, relative vorticity, stretching deformation, and shearing deformation. In Figure 13a, the zonal component shows westward winds in spring (March, April and May) and eastward winds in summer, autumn and winter (May to February) below 84 km, which is the characteristic of annual variation. However, within 84 to 90 km, the zonal wind shows maxima value in summer and winter as well as minimum value at the spring and autumn equinox, suggesting semiannual variation. From 90 to 96 km, the zonal wind again shows annual variation, with large eastward winds in spring to summer and low westward winds in autumn to winter. The overall zonal wind between 80 and 96 km is characterized by annual variation. During spring to summer, the zonal wind increases (greater eastward and lesser westward) with altitude, reaching a peak value of approximately 50 m/s at 96 km in May. In autumn and winter, the zonal wind decreases (lesser eastward and greater westward) with

altitude, reaching a minimum value at 96 km in November.

In Figure 13b, the meridional winds show similar structures to the zonal winds. Below 86 km, the meridional wind is southward in spring to summer and northward in autumn to winter. Meanwhile, at latitudes between 88 and 96 km, the meridional wind is northward in spring to summer and   is northward in autumn to winter, and the maximum values of northward and southward winds are 20 m/s and 15 m/s in May and October, respectively.

[Figure]

**Figure 13.** The seasonal values of horizontal winds and wind deformations from February 1, 2022 to February 1, 2023 at altitudes ranging from 80 to 96 km: (a) mean zonal wind, (b) mean meridional wind, (c) horizontal divergence, (d) relative vorticity,

(e) stretching deformation, and (f) shearing deformation.

In Figures 13c and 13d, the horizontal divergence and stretching deformation are calculated by the gradient terms $u_x$ and $v_y$, and they portray a similar seasonal variation. The divergence is almost negative below 84 km. Within 84 to 96 km, the value is still negative in March to June, reaching -0.1 m/s/km at 96 km in May. In the remaining months of the year, the divergence is positive, reaching 0.08 m/s/km at 94 km in November. The maximum values of divergence (convergence) correspond to the maxima of eastward/northward (westward/southward) wind velocities. For the stretching deformation, the result is totally negative below 88 km and shows the most negative value of -0.08 m/s/km at 80 km in January. In the upper altitude, the results are nearly negative in April to August and nearly positive in September to February, reaching 0.05 m/s/km at 96 km in February, which also corresponds to the maxima of the horizontal winds.

The relative vorticity in Figure 13e shows a clear stratification with height in summer and autumn, which is similar to the polar results reported by Chau et al. (2017). The value is negative between 84 and 92 km, with a maximum of -0.08 m/s/km in August at 88 km, and is positive below or above this height range. The maximum positive value is 0.18 m/s/km in January at 96 km. Finally, the wind shearing deformation, as shown in Figure 13e, is close to 0 and even negative near the spring and autumn equinox, which corresponds exactly to the annual variation structure of the background mean winds. The minimum and maximum values are both at 88 km, -0.04 m/s/km and 0.09 m/s/km, respectively, in March and December.

[Figure]

**Figure 14.** Height-time cross-sections of the zonal (a) and meridional (b) wind variance and the vertical flux of zonal (c) and meridional (d) momentum. Running averages over 30 days shifted by 7 days from February 2022 to February 2023. The black dashed lines indicate the zero mean zonal wind in (a) and (c) and zero mean meridional wind in (b) and (d).

Compared to monostatic meteor radar, more meteor detections and different viewing angles of the radial velocity observed by the bistatic meteor radar system can appreciably improve the precision of the GW covariance estimation (Spargo et al., 2019). In addition, we also try to use the bistatic meteor radar system to estimate the GW momentum fluxes, basically following the method introduced by Hocking (2005), Jia et al. (2018), Spargo et al. (2019) and the references therein. Figure 14 presents the seasonal variations in GW momentum flux terms.

As shown in Figures 14a and 14b, the zonal ($u'^2$) and meridional ($v'^2$) GW variance terms generally increase with height below ~90 km and then decrease with height above

90 km, reflecting the characteristics of the GW amplitude variation. The zonal GW variances indicate a semiannual variation, with two maxima around April and October and two minima in August and December.

The meridional ($v'^2$) GW variance mainly shows annual variation, with a maximum amplitude in winter and spring and a minimum amplitude in summer and autumn. Below 90 km, most gravity waves are unsaturated, so the amplitude of the gravity wave increases exponentially due to the decreasing air density, thus causing the GW variance to increase with height. Up to approximately 90 km, when the gravity wave is saturated, according to linear theory, the gravity wave will become unstable and break, leading to a decrease in the GW variance (Fritts, 1984; Placke et al., 2011).

In Figures 14c and 14d, the vertical fluxes of zonal (v'w') and meridional (u'w') momentum basically show a U-shape and an inverted U-shape, respectively. Both u'w' and v'w' decrease with height throughout the year, except for v'w' in September, when both reach maxima in September and minima in June but at different corresponding altitudes. Moreover, the meridional component appears to be southward during almost the whole year. These results are generally comparable with those of a previous study observed by Wuhan (30.2°N, 114.2°E) and Beijing (40°N, 116.3°E) at middle latitudes (e.g., Jia et al., 2018). Placke et al. (2011) reported that the GW activity in the upper mesosphere over Collm, Germany (51.31°N, 13.01°E) at higher-middle latitudes has a semiannual oscillation with the main maximum in summer and a minor maximum in winter. Combined with the background wind structure, we can see the influence of gravity waves on the structure of the background mean winds. From May to August, it can be considered that only the westward gravity wave can propagate upward under the background of the prevailing eastward wind, and the meanwhile, the westward momentum flux will produce the northward drag force under the action of the Coriolis force and thus slow down the southward wind.

**Track changes**

**Multistatic meteor radar observations of two-dimensional horizontal MLT wind**

Wen Yi [1,2], Jie Zeng[1,2], Xianghui Xue [1,2,3*], Iain Reid[4,5], Wei Zhong[1,2], Jianfei

Wu[1,2], Tingdi Chen[1,2], Xiankang Dou[1,6]

[1] Deep Space Exploration Laboratory / School of Earth and Space Sciences, University of Science and Technology of China, Hefei 230026, China
[2] CAS Key Laboratory of Geospace Environment/CAS Center for Excellence in Comparative Planetology, Anhui Mengcheng Geophysics National Observation and Research Station, University of Science and Technology of China, Hefei, China
[3] Collaborative Innovation Center of Astronautical Science and Technology, Harbin, China
[4] ATRAD Pty Ltd., Adelaide, SA 5000, Australia
[5] School of Physical Sciences, University of Adelaide, Adelaide, SA 5000, Australia
[6] Electronic Information School, Wuhan University, Wuhan, China

**Corresponding author:** Xianghui Xue (xuexh@ustc.edu.cn)

**Abstract:** All-sky meteor radars have become a reliable and widely used tool to observe horizontal winds in the mesosphere and lower thermosphere (MLT) region. The horizontal winds estimated by conventional single-station radars are obtained after averaging all meteor detections based on the assumption of the homogeneity of the horizontal wind in the meteor detection area (approximately 200-300 km radius). In this study, to improve the horizontal winds, we apply a multistatic meteor radar system consisting of a monostatic meteor radar in Mengcheng (33.36 °N, 116.49 °E) and a bistatic remote receiver in Changfeng (31.98 °N, 117.22 °E), separated by approximately 167 km to increase the number of meteors by at least 70% and provide two different viewing angles of the meteor echoes. The accuracy of the horizontal wind measurement depends on the meteor number in time and altitude intervals. Compared to typical monostatic meteor radar, our approach shows the feasibility of estimating the two-dimensional horizontal wind field. The technique allows us to estimate the mean horizontal wind and the gradient terms of the horizontal wind, as well as the horizontal divergence, relative vorticity, stretching and shearing deformation of the

wind field. In addition, the seasonal variation in MLT dynamics parameters, including horizontal wind deformations and gravity wave (GW) momentum fluxes, in the MLT region at middle latitudes of the Northern Hemisphere are presented. 
[revised manuscript text omitted]
., 2004). The composite error estimations are shown in Figure 12 (lower four rows) utilizing the data from October 16 to October 20, 2021. It is clear that the errors of horizontal winds and gradient terms are smaller than 1 m/s and 0.1 m/s/km, respectively, when the meteor detections are sufficient, such as the results ranging from 82 to 94 km during local morning (2000-0004 UT). The large errors basically occur above 94 km during local night (near 1200 UT), which is mainly caused by a low number of meteors.

Our results verify the ability of the VPP method to estimate  wind parameters. Based on these parameters, multistatic meteor radars are capable of deducing the inhomogeneities and kinetic characteristics of the wind fields, which are similar to those of Stober and Chau (2015). The increased meteor detections can reduce the error of the estimated terms and guarantee the reliability of the results.

**5. Seasonal variations dynamics parameters in MLT region**

This section uses the methodology described in the previous section to estimate the seasonal variation in MLT dynamics parameters, including horizontal wind deformations and gravity wave (GW) momentum fluxes, in the MLT region in the middle latitudes of the Northern Hemisphere from February 2022 to February 2023. As shown in Figure 13, we present the seasonal variations in monthly mean horizontal winds, horizontal divergence, relative vorticity, stretching deformation, and shearing

deformation. In Figure 13a, the zonal component shows westward winds in spring (March, April and May) and eastward winds in summer, autumn and winter (May to February) below 84 km, which is the characteristic of annual variation. However, within 84 to 90 km, the zonal wind shows maxima value in summer and winter as well as minimum value at the spring and autumn equinox, suggesting semiannual variation. From 90 to 96 km, the zonal wind again shows annual variation, with large eastward winds in spring to summer and low westward winds in autumn to winter. The overall zonal wind between 80 and 96 km is characterized by annual variation. During spring to summer, the zonal wind increases (greater eastward and lesser westward) with altitude, reaching a peak value of approximately 50 m/s at 96 km in May. In autumn and winter, the zonal wind decreases (lesser eastward and greater westward) with altitude, reaching a minimum value at 96 km in November.

In Figure 13b, the meridional winds show similar structures to the zonal winds. Below 86 km, the meridional wind is southward in spring to summer and northward in autumn to winter. Meanwhile, at latitudes between 88 and 96 km, the meridional wind is northward in spring to summer and  is northward in autumn to winter, and the maximum values of northward and southward winds are 20 m/s and 15 m/s in May and October, respectively.

[Figure]

**Figure 13.** The seasonal values of horizontal winds and wind deformations from February 1, 2022 to February 1, 2023 at altitudes ranging from 80 to 96 km: (a) mean zonal wind, (b) mean meridional wind, (c) horizontal divergence, (d) relative vorticity, (e) stretching deformation, and (f) shearing deformation.

In Figures 13c and 13d, the horizontal divergence and stretching deformation are calculated by the gradient terms $u_x$ and $v_y$, and they portray a similar seasonal variation. The divergence is almost negative below 84 km. Within 84 to 96 km, the value is still negative in March to June, reaching -0.1 m/s/km at 96 km in May. In the remaining months of the year, the divergence is positive, reaching 0.08 m/s/km at 94 km in November. The maximum values of divergence (convergence) correspond to the

maxima of eastward/northward (westward/southward) wind velocities. For the stretching deformation, the result is totally negative below 88 km and shows the most negative value of -0.08 m/s/km at 80 km in January. In the upper altitude, the results are nearly negative in April to August and nearly positive in September to February, reaching 0.05 m/s/km at 96 km in February, which also corresponds to the maxima of the horizontal winds.

The relative vorticity in Figure 13e shows a clear stratification with height in summer and autumn, which is similar to the polar results reported by Chau et al. (2017). The value is negative between 84 and 92 km, with a maximum of -0.08 m/s/km in August at 88 km, and is positive below or above this height range. The maximum positive value is 0.18 m/s/km in January at 96 km. Finally, the wind shearing deformation, as shown in Figure 13e, is close to 0 and even negative near the spring and autumn equinox, which corresponds exactly to the annual variation structure of the background mean winds. The minimum and maximum values are both at 88 km, -0.04 m/s/km and 0.09 m/s/km, respectively, in March and December.

[Figure]

**Figure 14.** Height-time cross-sections of the zonal (a) and meridional (b) wind variance and the vertical flux of zonal (c) and meridional (d) momentum. Running averages over 30 days shifted by 7 days from February 2022 to February 2023. The black dashed lines indicate the zero mean zonal wind in (a) and (c) and zero mean meridional wind in (b) and (d).

Compared to monostatic meteor radar, more meteor detections and different viewing angles of the radial velocity observed by the bistatic meteor radar system can appreciably improve the precision of the GW covariance estimation (Spargo et al., 2019). In addition, we also try to use the bistatic meteor radar system to estimate the GW momentum fluxes, basically following the method introduced by Hocking (2005), Jia et al. (2018), Spargo et al. (2019) and the references therein. Figure 14 presents the seasonal variations in GW momentum flux terms.

As shown in Figures 14a and 14b, the zonal ($u'^2$) and meridional ($v'^2$) GW variance terms generally increase with height below ~90 km and then decrease with height above 90 km, reflecting the characteristics of the GW amplitude variation. The zonal GW variances indicate a semiannual variation, with two maxima around April and October and two minima in August and December.

The meridional ($v'^2$) GW variance mainly shows annual variation, with a maximum amplitude in winter and spring and a minimum amplitude in summer and autumn. Below 90 km, most gravity waves are unsaturated, so the amplitude of the gravity wave increases exponentially due to the decreasing air density, thus causing the GW variance to increase with height. Up to approximately 90 km, when the gravity wave is saturated, according to linear theory, the gravity wave will become unstable and break, leading to a decrease in the GW variance (Fritts, 1984; Placke et al., 2011).

In Figures 14c and 14d, the vertical fluxes of zonal (v'w') and meridional (u'w') momentum basically show a U-shape and an inverted U-shape, respectively. Both u'w' and v'w' decrease with height throughout the year, except for v'w' in September, when both reach maxima in September and minima in June but at different corresponding altitudes. Moreover, the meridional component appears to be southward during almost the whole year. These results are generally comparable with those of a previous study observed by Wuhan (30.2°N, 114.2°E) and Beijing (40°N, 116.3°E) at middle latitudes (e.g., Jia et al., 2018). Placke et al. (2011) reported that the GW activity in the upper mesosphere over Collm, Germany (51.31°N, 13.01°E) at higher-middle latitudes has a semiannual oscillation with the main maximum in summer and a minor maximum in winter. Combined with the background wind structure, we can see the influence of gravity waves on the structure of the background mean winds. From May to August, it can be considered that only the westward gravity wave can propagate upward under the background of the prevailing eastward wind, and the meanwhile, the westward momentum flux will produce the northward drag force under the action of the Coriolis force and thus slow down the southward wind.

**6. Discussion and Summary**

In this study, we have presented  preliminary results from the Mengcheng and

Changfeng bistatic meteor radar systems. The main objectives were accomplished successfully by the new bistatic meteor radar system and are summarized as follows:

[revised manuscript text omitted]

---

## Author Comment (AC2)

We thank the reviewer for the useful suggestions to improve the paper. These comments are all valuable and very helpful for revising and improving our manuscript, as well as the important guiding significance to our research. These changes in the revised manuscript have been marked in the track changes version manuscript, as well as the point-to-point responses have been listed as follows:

**Response to reviewer #2**

Comment: This manuscript describes the use of multistatic meteor radars to study the mesosphere and lower thermosphere using installations in central China to illustrate their methods. It follows the structure of other similar papers and does not bring new content to the field other than a new location. (It was much less clear than its predecessors with missing variable definitions and undescribed terms.) As such, it does not warrant publication in AMT.

**Response:** Thank you for your great suggestion. We agree with the shortcomings of our study, especially the innovation of methods/algorithms and a few days of observation. To provide more new information based on the bistatic meteor radar system, we extended the observation data to more than 1 year and added a new section in the revised manuscript. This study will provide the first investigation of the seasonal variations in MLT dynamics at lower-midlatitudes (near 30°N) in the Northern Hemisphere.

*Here is what we have added in the revised manuscript.*

**5. Seasonal variation dynamics parameters in the MLT region**

This section uses the methodology described in the previous section to estimate the seasonal variation in MLT dynamics parameters, including horizontal wind deformations and gravity wave (GW) momentum fluxes, in the MLT region in the middle latitudes of the Northern Hemisphere from February 2022 to February 2023. As shown in Figure 13, we present the seasonal variations in monthly mean horizontal winds, horizontal divergence, relative vorticity, stretching deformation, and shearing deformation. In Figure 13a, the zonal component shows westward winds in spring

(March, April and May) and eastward winds in summer, autumn and winter (May to February) below 84 km, which is the characteristic of annual variation. However, within 84 to 90 km, the zonal wind shows maxima value in summer and winter as well as minimum value at the spring and autumn equinox, suggesting semiannual variation. From 90 to 96 km, the zonal wind again shows annual variation, with large eastward winds in spring to summer and low westward winds in autumn to winter. The overall zonal wind between 80 and 96 km is characterized by annual variation. During spring to summer, the zonal wind increases (greater eastward and lesser westward) with altitude, reaching a peak value of approximately 50 m/s at 96 km in May. In autumn and winter, the zonal wind decreases (lesser eastward and greater westward) with altitude, reaching a minimum value at 96 km in November.

In Figure 13b, the meridional winds show similar structures to the zonal winds. Below 86 km, the meridional wind is southward in spring to summer and northward in autumn to winter. Meanwhile, at latitudes between 88 and 96 km, the meridional wind is northward in spring to summer and  is northward in autumn to winter, and the maximum values of northward and southward winds are 20 m/s and 15 m/s in May and October, respectively.

[Figure]

**Figure 13.** The seasonal values of horizontal winds and wind deformations from February 1, 2022 to February 1, 2023 at altitudes ranging from 80 to 96 km: (a) mean zonal wind, (b) mean meridional wind, (c) horizontal divergence, (d) relative vorticity, (e) stretching deformation, and (f) shearing deformation.

In Figures 13c and 13d, the horizontal divergence and stretching deformation are calculated by the gradient terms $u_x$ and $v_y$, and they portray a similar seasonal variation. The divergence is almost negative below 84 km. Within 84 to 96 km, the value is still negative in March to June, reaching -0.1 m/s/km at 96 km in May. In the remaining months of the year, the divergence is positive, reaching 0.08 m/s/km at 94 km in November. The maximum values of divergence (convergence) correspond to the

maxima of eastward/northward (westward/southward) wind velocities. For the stretching deformation, the result is totally negative below 88 km and shows the most negative value of -0.08 m/s/km at 80 km in January. In the upper altitude, the results are nearly negative in April to August and nearly positive in September to February, reaching 0.05 m/s/km at 96 km in February, which also corresponds to the maxima of the horizontal winds.

The relative vorticity in Figure 13e shows a clear stratification with height in summer and autumn, which is similar to the polar results reported by Chau et al. (2017). The value is negative between 84 and 92 km, with a maximum of -0.08 m/s/km in August at 88 km, and is positive below or above this height range. The maximum positive value is 0.18 m/s/km in January at 96 km. Finally, the wind shearing deformation, as shown in Figure 13e, is close to 0 and even negative near the spring and autumn equinox, which corresponds exactly to the annual variation structure of the background mean winds. The minimum and maximum values are both at 88 km, -0.04 m/s/km and 0.09 m/s/km, respectively, in March and December.

[Figure]

**Figure 14.** Height-time cross-sections of the zonal (a) and meridional (b) wind variance and the vertical flux of zonal (c) and meridional (d) momentum. Running averages over 30 days shifted by 7 days from February 2022 to February 2023. The black dashed lines indicate the zero mean zonal wind in (a) and (c) and zero mean meridional wind in (b) and (d).

Compared to monostatic meteor radar, more meteor detections and different viewing angles of the radial velocity observed by the bistatic meteor radar system can appreciably improve the precision of the GW covariance estimation (Spargo et al., 2019). In addition, we also try to use the bistatic meteor radar system to estimate the GW momentum fluxes, basically following the method introduced by Hocking (2005), Jia et al. (2018), Spargo et al. (2019) and the references therein. Figure 14 presents the seasonal variations in GW momentum flux terms.

As shown in Figures 14a and 14b, the zonal ($u'^2$) and meridional ($v'^2$) GW variance

terms generally increase with height below ~90 km and then decrease with height above 90 km, reflecting the characteristics of the GW amplitude variation. The zonal GW variances indicate a semiannual variation, with two maxima around April and October and two minima in August and December.

The meridional ($v'^2$) GW variance mainly shows annual variation, with a maximum amplitude in winter and spring and a minimum amplitude in summer and autumn. Below 90 km, most gravity waves are unsaturated, so the amplitude of the gravity wave increases exponentially due to the decreasing air density, thus causing the GW variance to increase with height. Up to approximately 90 km, when the gravity wave is saturated, according to linear theory, the gravity wave will become unstable and break, leading to a decrease in the GW variance (Fritts, 1984; Placke et al., 2011).

In Figures 14c and 14d, the vertical fluxes of zonal (v'w') and meridional (u'w') momentum basically show a U-shape and an inverted U-shape, respectively. Both u'w' and v'w' decrease with height throughout the year, except for v'w' in September, when both reach maxima in September and minima in June but at different corresponding altitudes. Moreover, the meridional component appears to be southward during almost the whole year. These results are generally comparable with those of a previous study observed by Wuhan (30.2°N, 114.2°E) and Beijing (40°N, 116.3°E) at middle latitudes (e.g., Jia et al., 2018). Placke et al. (2011) reported that the GW activity in the upper mesosphere over Collm, Germany (51.31°N, 13.01°E) at higher-middle latitudes has a semiannual oscillation with the main maximum in summer and a minor maximum in winter. Combined with the background wind structure, we can see the influence of gravity waves on the structure of the background mean winds. From May to August, it can be considered that only the westward gravity wave can propagate upward under the background of the prevailing eastward wind, and the meanwhile, the westward momentum flux will produce the northward drag force under the action of the Coriolis force and thus slow down the southward wind.

**Comment:** Some aspects of the paper make me concerned that the analysis as presented has merit or was correctly applied. It is assumed that the vertical velocity can be ignored

in equations (7) and (8) without adequate justification.

**Response:** We have added the vertical velocity, and here, we focus only on the horizontal results, so we set the vertical velocity $w = w_0$. Many previous studies have estimated that the magnitude of the vertical velocity is just 0.1 cm/s to several m/s, which is far below the magnitude of the horizontal wind velocities (e.g., Vincent et al., 2019; Stober et al., 2022, and references therein). However, there were also some attempts to fit vertical winds to the observations (e.g., Chau et al., 2017; Conte et al., 2021; Chau et al., 2021), and the results may be contaminated by horizontal terms. Thereafter, we should note that the resulting vertical velocity presents not the real vertical wind but the so-called apparent vertical wind ($w^*$) (Chau et al., 2017). The vertical velocity results of a few days and a year are shown in Figure R1. The estimated $w^*$ presents a diurnal variation in Figure R1 (a) and shows a similar annual variation to that of the horizontal winds.

After revising equations (7) and (8), we have changed the corresponding contents of Figure 7 – Figure 12 in the revised manuscript. Although the overall change is small, we still find that the divergence decreases, which may lead to the large vertical velocity shown in Figure R1 (Chau et al., 2017).

[Figure]

**Figure R1.** The vertical velocity of (a) October 16 and 17, 2021, and (b) February 1, 2022, to February 1, 2023. The thin black dashed lines in each panel present the equinoxes.

In the next stage, we will expand our multistatic meteor radar network in Central-Eastern China. The new multistatic meteor radar network will provide a better determination of vertical winds refer to the more recent results reported by Stober et al., (2021; 2022).

**Comment:** Differences in the winds portayed in Figure 6 approach 40 m/sec at heights where the meteor count rates should be large. Such large discrepancies between locations that are not far apart will need to be explained before readers can accept the more nuanced parameters describing velocity variations within the field.

The opportunities that will be afforded by the construction of a multi-radar array in central China are considerable but the above aspects of the analysis need to be resolved to ensure the results reflect true geophysical effects.

**Response:** The large discrepancies in Figure 6 may be generated for a few reasons. First, there are unevenly distributed meteors. Taking a look at Figure 5, we find that the distributions of the arrival angles of the two receiving sites are not totally the same. We select the angle data corresponding to the bins with sufficient meteors, compare these data of the two sites, and then find that most of the bins with large discrepancies in wind velocity have inconsistent angle distributions. Second, the distance between the two sites is 167.3 km, but the valid detection range of the Mengcheng site is just 150 km, so the observed fields are not the same. Once affected by strong atmospheric motions, it is possible to have large discrepancies. Third, most of the large discrepancies exist at 12 LT at lower/upper altitudes, and the meteors in this range are not sufficient to estimate accurate horizontal winds. Due to the combined influence of the above several factors, we believe our results are reasonable.

**References in this response:**

Chau, J. L., G. Stober, C. M. Hall, M. Tsutsumi, F. I. Laskar, and P. Hoffmann (2017), Polar mesospheric horizontal divergence and relative vorticity measurements using multiple specular meteor radars, Radio Sci., 52, 811–828,

doi:10.1002/2016RS006225. Stober, G., Chau, J. L., Vierinen, J., Jacobi, C., and Wilhelm, S.: Retrieving horizontally resolved wind fields using multi-static meteor radar observations, Atmos. Meas. Tech., 11, 4891–4907, https://doi.org/10.5194/amt-11-4891-2018, 2018.

Chau, J. L., Urco, J. M., Vierinen, J., Harding, B. J., Clahsen, M., Pfeffer, N., et al. (2021). Multistatic specular meteor radar network in Peru: System description and initial results. Earth and Space Science, 8, e2020EA001293. https://doi.org/10.1029/2020EA001293.

Conte, J. F., Chau, J. L., Urco, J. M., Latteck, R., Vierinen, J., and Salvador, J. O.: First Studies of Mesosphere and Lower Thermosphere Dynamics Using a Multistatic Specular Meteor Radar Network Over Southern Patagonia, Earth and Space Science, 8, e2020EA001356, https://doi.org/10.1029/2020EA001356, 2021

Poblet, F. L., Chau, J. L., Conte, J. F., Avsarkisov, V., Vierinen, J., & Charuvil Asokan, H. (2022). Horizontal wavenumber spectra of vertical vorticity and horizontal divergence of mesoscale dynamics in the mesosphere and lower thermosphere using multistatic specular meteor radar observations. Earth and Space Science, 9, e2021EA002201. https://doi.org/10.1029/2021EA002201.

Spargo, A. J., Reid, I. M., and MacKinnon, A. D.: Multistatic meteor radar observations of gravity-wave–tidal interaction over southern Australia, Atmos. Meas. Tech., 12, 4791–4812, https://doi.org/10.5194/amt-12-4791-2019, 2019.

Stober, G., and J. L. Chau (2015), A multistatic and multifrequency novel approach for specular meteor radars to improve wind measurements in the MLT region, Radio Sci., 50, 431–442, doi:10.1002/2014RS005591.

Vierinen, J., Chau, J. L., Charuvil, H., Urco, J. M., Clahsen, M., Avsarkisov, V., et al. (2019). Observing mesospheric turbulence with specular meteor radars: A novel method for estimating second-order statistics of wind velocity. Earth and Space Science, 6. https://doi.org/10.1029/2019EA000570.

Vincent, R. A., Kovalam, S., Murphy, D. J., Reid, I. M., and Younger, J. P.: Trends and Variability in Vertical Winds in the Southern Hemisphere Summer Polar Mesosphere and Lower Thermosphere, J. Geophys. Res.-Atmos., 124, 11070–

11085, https://doi.org/10.1029/2019JD030735, 2019

**Tracked Changes**

**Multistatic meteor radar observations of two-dimensional horizontal MLT wind**

Wen Yi [1,2], Jie Zeng[1,2], Xianghui Xue [1,2,3*], Iain Reid[4,5], Wei Zhong[1,2], Jianfei Wu[1,2], Tingdi Chen[1,2], Xiankang Dou[1,6]

[1] Deep Space Exploration Laboratory / School of Earth and Space Sciences, University of Science and Technology of China, Hefei 230026, China

[2] CAS Key Laboratory of Geospace Environment/CAS Center for Excellence in Comparative Planetology, Anhui Mengcheng Geophysics National Observation and Research Station, University of Science and Technology of China, Hefei, China

[3] Collaborative Innovation Center of Astronautical Science and Technology, Harbin, China

[4] ATRAD Pty Ltd., Adelaide, SA 5000, Australia

[5] School of Physical Sciences, University of Adelaide, Adelaide, SA 5000, Australia

[6] Electronic Information School, Wuhan University, Wuhan, China

**Corresponding author:** Xianghui Xue (xuexh@ustc.edu.cn)

**Abstract:** All-sky meteor radars have become a reliable and widely used tool to observe horizontal winds in the mesosphere and lower thermosphere (MLT) region. The horizontal winds estimated by conventional single-station radars are obtained after averaging all meteor detections based on the assumption of the homogeneity of the horizontal wind in the meteor detection area (approximately 200-300 km radius). In this study, to improve the horizontal winds, we apply a multistatic meteor radar system consisting of a monostatic meteor radar in Mengcheng (33.36 °N, 116.49 °E) and a bistatic remote receiver in Changfeng (31.98 °N, 117.22 °E), separated by approximately 167 km to increase the number of meteors by at least 70% and provide

two different viewing angles of the meteor echoes. The accuracy of the horizontal wind measurement depends on the meteor number in time and altitude intervals. Compared to typical monostatic meteor radar, our approach shows the feasibility of estimating the two-dimensional horizontal wind field. The technique allows us to estimate the mean horizontal wind and the gradient terms of the horizontal wind, as well as the horizontal divergence, relative vorticity, stretching and shearing deformation of the wind field. In addition, the seasonal variation in MLT dynamics parameters, including horizontal wind deformations and gravity wave (GW) momentum fluxes, in the MLT region at middle latitudes of the Northern Hemisphere are presented. 
[revised manuscript text omitted]
., 2004). The composite error estimations are shown in Figure 12 (lower four rows) utilizing the data from October 16 to October 20, 2021. It is clear that the errors of horizontal winds and gradient terms are smaller than 1 m/s and 0.1 m/s/km, respectively, when the meteor detections are sufficient, such as the results ranging from 82 to 94 km during local morning (2000-0004 UT). The large errors basically occur above 94 km during local night (near 1200 UT), which is mainly caused by a low number of meteors.

Our results verify the ability of the VPP method to estimate  wind parameters. Based on these parameters, multistatic meteor radars are capable of deducing the inhomogeneities and kinetic characteristics of the wind fields, which are similar to those of Stober and Chau (2015). The increased meteor detections can reduce the error of the estimated terms and guarantee the reliability of the results.

**5. Seasonal variations dynamics parameters in MLT region**

This section uses the methodology described in the previous section to estimate the seasonal variation in MLT dynamics parameters, including horizontal wind deformations and gravity wave (GW) momentum fluxes, in the MLT region in the middle latitudes of the Northern Hemisphere from February 2022 to February 2023. As shown in Figure 13, we present the seasonal variations in monthly mean horizontal winds, horizontal divergence, relative vorticity, stretching deformation, and shearing

deformation. In Figure 13a, the zonal component shows westward winds in spring (March, April and May) and eastward winds in summer, autumn and winter (May to February) below 84 km, which is the characteristic of annual variation. However, within 84 to 90 km, the zonal wind shows maxima value in summer and winter as well as minimum value at the spring and autumn equinox, suggesting semiannual variation. From 90 to 96 km, the zonal wind again shows annual variation, with large eastward winds in spring to summer and low westward winds in autumn to winter. The overall zonal wind between 80 and 96 km is characterized by annual variation. During spring to summer, the zonal wind increases (greater eastward and lesser westward) with altitude, reaching a peak value of approximately 50 m/s at 96 km in May. In autumn and winter, the zonal wind decreases (lesser eastward and greater westward) with altitude, reaching a minimum value at 96 km in November.

In Figure 13b, the meridional winds show similar structures to the zonal winds. Below 86 km, the meridional wind is southward in spring to summer and northward in autumn to winter. Meanwhile, at latitudes between 88 and 96 km, the meridional wind is northward in spring to summer and  is northward in autumn to winter, and the maximum values of northward and southward winds are 20 m/s and 15 m/s in May and October, respectively.

[Figure]

**Figure 13.** The seasonal values of horizontal winds and wind deformations from February 1, 2022 to February 1, 2023 at altitudes ranging from 80 to 96 km: (a) mean zonal wind, (b) mean meridional wind, (c) horizontal divergence, (d) relative vorticity, (e) stretching deformation, and (f) shearing deformation.

In Figures 13c and 13d, the horizontal divergence and stretching deformation are calculated by the gradient terms $u_x$ and $v_y$, and they portray a similar seasonal variation. The divergence is almost negative below 84 km. Within 84 to 96 km, the value is still negative in March to June, reaching -0.1 m/s/km at 96 km in May. In the remaining months of the year, the divergence is positive, reaching 0.08 m/s/km at 94 km in November. The maximum values of divergence (convergence) correspond to the

maxima of eastward/northward (westward/southward) wind velocities. For the stretching deformation, the result is totally negative below 88 km and shows the most negative value of -0.08 m/s/km at 80 km in January. In the upper altitude, the results are nearly negative in April to August and nearly positive in September to February, reaching 0.05 m/s/km at 96 km in February, which also corresponds to the maxima of the horizontal winds.

The relative vorticity in Figure 13e shows a clear stratification with height in summer and autumn, which is similar to the polar results reported by Chau et al. (2017). The value is negative between 84 and 92 km, with a maximum of -0.08 m/s/km in August at 88 km, and is positive below or above this height range. The maximum positive value is 0.18 m/s/km in January at 96 km. Finally, the wind shearing deformation, as shown in Figure 13e, is close to 0 and even negative near the spring and autumn equinox, which corresponds exactly to the annual variation structure of the background mean winds. The minimum and maximum values are both at 88 km, -0.04 m/s/km and 0.09 m/s/km, respectively, in March and December.

[Figure]

**Figure 14.** Height-time cross-sections of the zonal (a) and meridional (b) wind variance and the vertical flux of zonal (c) and meridional (d) momentum. Running averages over 30 days shifted by 7 days from February 2022 to February 2023. The black dashed lines indicate the zero mean zonal wind in (a) and (c) and zero mean meridional wind in (b) and (d).

Compared to monostatic meteor radar, more meteor detections and different viewing angles of the radial velocity observed by the bistatic meteor radar system can appreciably improve the precision of the GW covariance estimation (Spargo et al., 2019). In addition, we also try to use the bistatic meteor radar system to estimate the GW momentum fluxes, basically following the method introduced by Hocking (2005), Jia et al. (2018), Spargo et al. (2019) and the references therein. Figure 14 presents the seasonal variations in GW momentum flux terms.

As shown in Figures 14a and 14b, the zonal ($u'^2$) and meridional ($v'^2$) GW variance terms generally increase with height below ~90 km and then decrease with height above 90 km, reflecting the characteristics of the GW amplitude variation. The zonal GW variances indicate a semiannual variation, with two maxima around April and October and two minima in August and December.

The meridional ($v'^2$) GW variance mainly shows annual variation, with a maximum amplitude in winter and spring and a minimum amplitude in summer and autumn. Below 90 km, most gravity waves are unsaturated, so the amplitude of the gravity wave increases exponentially due to the decreasing air density, thus causing the GW variance to increase with height. Up to approximately 90 km, when the gravity wave is saturated, according to linear theory, the gravity wave will become unstable and break, leading to a decrease in the GW variance (Fritts, 1984; Placke et al., 2011).

In Figures 14c and 14d, the vertical fluxes of zonal (v'w') and meridional (u'w') momentum basically show a U-shape and an inverted U-shape, respectively. Both u'w' and v'w' decrease with height throughout the year, except for v'w' in September, when both reach maxima in September and minima in June but at different corresponding altitudes. Moreover, the meridional component appears to be southward during almost the whole year. These results are generally comparable with those of a previous study observed by Wuhan (30.2°N, 114.2°E) and Beijing (40°N, 116.3°E) at middle latitudes (e.g., Jia et al., 2018). Placke et al. (2011) reported that the GW activity in the upper mesosphere over Collm, Germany (51.31°N, 13.01°E) at higher-middle latitudes has a semiannual oscillation with the main maximum in summer and a minor maximum in winter. Combined with the background wind structure, we can see the influence of gravity waves on the structure of the background mean winds. From May to August, it can be considered that only the westward gravity wave can propagate upward under the background of the prevailing eastward wind, and the meanwhile, the westward momentum flux will produce the northward drag force under the action of the Coriolis force and thus slow down the southward wind.

**6. Discussion and Summary**

In this study, we have presented  preliminary results from the Mengcheng and

Changfeng bistatic meteor radar systems. The main objectives were accomplished successfully by the new bistatic meteor radar system and are summarized as follows:

[revised manuscript text omitted]

---

## Author Comment (AC3)

We thank the reviewer for the useful suggestions to improve the paper. These comments are all valuable and very helpful for revising and improving our manuscript, as well as the important guiding significance to our research. These changes in the revised manuscript have been marked in the track changes version manuscript, as well as the point-to-point responses have been listed as follows:

**Response to reviewer #1**

**General comments:** This manuscript describes a bi-static meteor radar set up in China and reported preliminary analysis that demonstrates the ability of this system in inferring horizontal wind variations. The manuscript is clearly written and easy to follow, and the graphics are nicely prepared. While there are already several multistatic meteor radars around the world and they are being reported as pointed out by another reviewer, a new system in China is still worth reporting.

However, given the knowledge already learned from other systems, this work needs to focus on something new about this system, rather than only presenting a few days' results using one of the same processing methods.

**Response:** Thank you for your valuable suggestion. We have added a section introducing seasonal variations in some MLT dynamics parameters, and these parameters include the horizontal mean winds, wind deformations, and gravity wave momentum flux. Since we will conduct further in-depth analysis on these parameters, only preliminary results are presented in the revised manuscript.

**Comment:** A few important questions could be addressed that are not yet adequately addressed in current publications. For example, do the tidal components of the divergence and vorticity consistent with the theory?

**Response:** Currently, there is little research concerning divergence and vorticity, but our results are consistent with those of previous studies. Chau et al. (2017) found semidiurnal signatures in the gradient terms, which were similar to the horizontal mean winds in polar areas. Conte et al. (2020) also found that structures in gradient terms

were similar to those in mean winds and suggested that they are influenced by tides and gravity waves. However, limited by measurement techniques and the existing research, physical interpretations of the derived terms will be left in our further studies.

Comment: How to reconcile the divergence field with the zero vertical wind assumption? The divergence is quite strong, which implies a strong vertical wind.

**Response:** We have revised equations (7) – (8) and added the vertical velocity to the calculation, taking the approach from Stober and Chau (2015). The resulting divergence value decreases due to the recalculation. However, the resulting vertical velocity reaches a few m/s, which is more similar to the apparent vertical velocity due to horizontal divergence contamination (Chau et al., 2017). Vincent et al. (2019) estimated the polar vertical velocity using the divergence of the mean meridional wind using wind measurements of more than 20 years, and the resulting magnitude was several cm/s. Stober et al. (2022) introduced a 3DVAR+DIV algorithm to estimate the real vertical velocity, and they obtained vertical velocities in the range of $\pm1$–2 m/s for most of the analysed data during 2 years of collection. Due to the lack of direct measurement and the difficulty of estimation, the algorithm for vertical velocity estimation will be the focus of our next work.

References in this response:

Chau J L, Stober G, Hall C M, Tsutsumi M, Laskar F I, Hoffmann P. 2017. Polar mesospheric horizontal divergence and relative vorticity measurements using multiple specular meteor radars. *Radio Science*, *52*, 811–828. https://doi.org/10.1002/2016RS006225

Conte J F, Chau J L, Urco J M, Latteck R, Vierinen J, Salvador J O. 2021. First studies of mesosphere and lower thermosphere dynamics using a multistatic specular meteor radar network over southern Patagonia. *Earth and Space Science*, *8*, e2020EA001356. https://doi.org/10.1029/2020EA001356

Stober G, Chau J. (2015). A multistatic and multifrequency novel approach for specular meteor radars to improve wind measurements in the MLT region. *Radio Science*, *50*, 431–442. https://doi.org/10.1002/2014rs005591

Stober, G., Liu, A., Kozlovsky, A., Qiao, Z., Kuchar, A., Jacobi, C., Meek, C., Janches, D., Liu, G., Tsutsumi, M., Gulbrandsen, N., Nozawa, S., Lester, M., Belova, E., Kero, J., and Mitchell, N. (2022) Meteor radar vertical wind observation biases

and mathematical debiasing strategies including the 3DVAR+DIV algorithm, *Atmos. Meas. Tech.*, 15, 5769–5792, https://doi.org/10.5194/amt-15-5769-2022.

Vincent, R. A., Kovalam, S., Murphy, D. J., Reid, I. M., and Younger, J. P.: Trends and Variability in Vertical Winds in the Southern Hemisphere Summer Polar Mesosphere and Lower Thermosphere, J. Geophys. Res.-Atmos., 124, 11070–11085, https://doi.org/10.1029/2019JD030735, 2019

**Tracked changes**

**Multistatic meteor radar observations of two-dimensional horizontal MLT wind**

Wen Yi [1,2], Jie Zeng[1,2], Xianghui Xue [1,2,3*], Iain Reid[4,5], Wei Zhong[1,2], Jianfei Wu[1,2], Tingdi Chen[1,2], Xiankang Dou[1,6]

[1] Deep Space Exploration Laboratory / School of Earth and Space Sciences, University of Science and Technology of China, Hefei 230026, China

[2] CAS Key Laboratory of Geospace Environment/CAS Center for Excellence in Comparative Planetology, Anhui Mengcheng Geophysics National Observation and Research Station, University of Science and Technology of China, Hefei, China

[3] Collaborative Innovation Center of Astronautical Science and Technology, Harbin, China

[4] ATRAD Pty Ltd., Adelaide, SA 5000, Australia

[5] School of Physical Sciences, University of Adelaide, Adelaide, SA 5000, Australia

[6] Electronic Information School, Wuhan University, Wuhan, China

**Corresponding author:** Xianghui Xue (xuexh@ustc.edu.cn)

**Abstract:** All-sky meteor radars have become a reliable and widely used tool to observe horizontal winds in the mesosphere and lower thermosphere (MLT) region. The horizontal winds estimated by conventional single-station radars are obtained after averaging all meteor detections based on the assumption of the homogeneity of the horizontal wind in the meteor detection area (approximately 200-300 km radius). In this study, to improve the horizontal winds, we apply a multistatic meteor radar system consisting of a monostatic meteor radar in Mengcheng (33.36 °N, 116.49 °E) and a bistatic remote receiver in Changfeng (31.98 °N, 117.22 °E), separated by

approximately 167 km to increase the number of meteors by at least 70% and provide two different viewing angles of the meteor echoes. The accuracy of the horizontal wind measurement depends on the meteor number in time and altitude intervals. Compared to typical monostatic meteor radar, our approach shows the feasibility of estimating the two-dimensional horizontal wind field. The technique allows us to estimate the mean horizontal wind and the gradient terms of the horizontal wind, as well as the horizontal divergence, relative vorticity, stretching and shearing deformation of the wind field. In addition, the seasonal variation in MLT dynamics parameters, including horizontal wind deformations and gravity wave (GW) momentum fluxes, in the MLT region at middle latitudes of the Northern Hemisphere are presented. 
[revised manuscript text omitted]
., 2004). The composite error estimations are shown in Figure 12 (lower four rows) utilizing the data from October 16 to October 20, 2021. It is clear that the errors of horizontal winds and gradient terms are smaller than 1 m/s and 0.1 m/s/km, respectively, when the meteor detections are sufficient, such as the results ranging from 82 to 94 km during local morning (2000-0004 UT). The large errors basically occur above 94 km during local night (near 1200 UT), which is mainly caused by a low number of meteors.

Our results verify the ability of the VPP method to estimate  wind parameters. Based on these parameters, multistatic meteor radars are capable of deducing the inhomogeneities and kinetic characteristics of the wind fields, which are similar to those of Stober and Chau (2015). The increased meteor detections can reduce the error of the estimated terms and guarantee the reliability of the results.

**5. Seasonal variations dynamics parameters in MLT region**

This section uses the methodology described in the previous section to estimate the seasonal variation in MLT dynamics parameters, including horizontal wind deformations and gravity wave (GW) momentum fluxes, in the MLT region in the middle latitudes of the Northern Hemisphere from February 2022 to February 2023. As shown in Figure 13, we present the seasonal variations in monthly mean horizontal winds, horizontal divergence, relative vorticity, stretching deformation, and shearing

deformation. In Figure 13a, the zonal component shows westward winds in spring (March, April and May) and eastward winds in summer, autumn and winter (May to February) below 84 km, which is the characteristic of annual variation. However, within 84 to 90 km, the zonal wind shows maxima value in summer and winter as well as minimum value at the spring and autumn equinox, suggesting semiannual variation. From 90 to 96 km, the zonal wind again shows annual variation, with large eastward winds in spring to summer and low westward winds in autumn to winter. The overall zonal wind between 80 and 96 km is characterized by annual variation. During spring to summer, the zonal wind increases (greater eastward and lesser westward) with altitude, reaching a peak value of approximately 50 m/s at 96 km in May. In autumn and winter, the zonal wind decreases (lesser eastward and greater westward) with altitude, reaching a minimum value at 96 km in November.

In Figure 13b, the meridional winds show similar structures to the zonal winds. Below 86 km, the meridional wind is southward in spring to summer and northward in autumn to winter. Meanwhile, at latitudes between 88 and 96 km, the meridional wind is northward in spring to summer and  is northward in autumn to winter, and the maximum values of northward and southward winds are 20 m/s and 15 m/s in May and October, respectively.

[Figure]

**Figure 13.** The seasonal values of horizontal winds and wind deformations from February 1, 2022 to February 1, 2023 at altitudes ranging from 80 to 96 km: (a) mean zonal wind, (b) mean meridional wind, (c) horizontal divergence, (d) relative vorticity, (e) stretching deformation, and (f) shearing deformation.

In Figures 13c and 13d, the horizontal divergence and stretching deformation are calculated by the gradient terms $u_x$ and $v_y$, and they portray a similar seasonal variation. The divergence is almost negative below 84 km. Within 84 to 96 km, the value is still negative in March to June, reaching -0.1 m/s/km at 96 km in May. In the remaining months of the year, the divergence is positive, reaching 0.08 m/s/km at 94 km in November. The maximum values of divergence (convergence) correspond to the

maxima of eastward/northward (westward/southward) wind velocities. For the stretching deformation, the result is totally negative below 88 km and shows the most negative value of -0.08 m/s/km at 80 km in January. In the upper altitude, the results are nearly negative in April to August and nearly positive in September to February, reaching 0.05 m/s/km at 96 km in February, which also corresponds to the maxima of the horizontal winds.

The relative vorticity in Figure 13e shows a clear stratification with height in summer and autumn, which is similar to the polar results reported by Chau et al. (2017). The value is negative between 84 and 92 km, with a maximum of -0.08 m/s/km in August at 88 km, and is positive below or above this height range. The maximum positive value is 0.18 m/s/km in January at 96 km. Finally, the wind shearing deformation, as shown in Figure 13e, is close to 0 and even negative near the spring and autumn equinox, which corresponds exactly to the annual variation structure of the background mean winds. The minimum and maximum values are both at 88 km, -0.04 m/s/km and 0.09 m/s/km, respectively, in March and December.

[Figure]

**Figure 14.** Height-time cross-sections of the zonal (a) and meridional (b) wind variance and the vertical flux of zonal (c) and meridional (d) momentum. Running averages over 30 days shifted by 7 days from February 2022 to February 2023. The black dashed lines indicate the zero mean zonal wind in (a) and (c) and zero mean meridional wind in (b) and (d).

Compared to monostatic meteor radar, more meteor detections and different viewing angles of the radial velocity observed by the bistatic meteor radar system can appreciably improve the precision of the GW covariance estimation (Spargo et al., 2019). In addition, we also try to use the bistatic meteor radar system to estimate the GW momentum fluxes, basically following the method introduced by Hocking (2005), Jia et al. (2018), Spargo et al. (2019) and the references therein. Figure 14 presents the seasonal variations in GW momentum flux terms.

As shown in Figures 14a and 14b, the zonal ($u'^2$) and meridional ($v'^2$) GW variance terms generally increase with height below ~90 km and then decrease with height above 90 km, reflecting the characteristics of the GW amplitude variation. The zonal GW variances indicate a semiannual variation, with two maxima around April and October and two minima in August and December.

The meridional ($v'^2$) GW variance mainly shows annual variation, with a maximum amplitude in winter and spring and a minimum amplitude in summer and autumn. Below 90 km, most gravity waves are unsaturated, so the amplitude of the gravity wave increases exponentially due to the decreasing air density, thus causing the GW variance to increase with height. Up to approximately 90 km, when the gravity wave is saturated, according to linear theory, the gravity wave will become unstable and break, leading to a decrease in the GW variance (Fritts, 1984; Placke et al., 2011).

In Figures 14c and 14d, the vertical fluxes of zonal (v'w') and meridional (u'w') momentum basically show a U-shape and an inverted U-shape, respectively. Both u'w' and v'w' decrease with height throughout the year, except for v'w' in September, when both reach maxima in September and minima in June but at different corresponding altitudes. Moreover, the meridional component appears to be southward during almost the whole year. These results are generally comparable with those of a previous study observed by Wuhan (30.2°N, 114.2°E) and Beijing (40°N, 116.3°E) at middle latitudes (e.g., Jia et al., 2018). Placke et al. (2011) reported that the GW activity in the upper mesosphere over Collm, Germany (51.31°N, 13.01°E) at higher-middle latitudes has a semiannual oscillation with the main maximum in summer and a minor maximum in winter. Combined with the background wind structure, we can see the influence of gravity waves on the structure of the background mean winds. From May to August, it can be considered that only the westward gravity wave can propagate upward under the background of the prevailing eastward wind, and the meanwhile, the westward momentum flux will produce the northward drag force under the action of the Coriolis force and thus slow down the southward wind.

**6. Discussion and Summary**

In this study, we have presented  preliminary results from the Mengcheng and

Changfeng bistatic meteor radar systems. The main objectives were accomplished successfully by the new bistatic meteor radar system and are summarized as follows:

[revised manuscript text omitted]

---

## Author Comment (AC4)

We thank the reviewer for the useful suggestions to improve the paper. These comments are all valuable and very helpful for revising and improving our manuscript, as well as the important guiding significance to our research. These changes in the revised manuscript have been marked in the track changes version manuscript, as well as the point-to-point responses have been listed as follows:

**Response to reviewer #3**

**Comments:**

This paper, "Multistatic meteor radar observations of two-dimensional horizontal MLT wind", introduces a bistatic extension of a traditional all-sky meteor radar system in an effort to estimate two-dimensional horizontal wind fields by relaxing the usual wind field spatial homogeneity assumption. The authors provide a good historical context for the work, and highlight recent developments in the field. The work is generally well organized, but could use additional grammar review. The manuscript meets basic scientific quality, is free from obvious major deficiencies, and is suitable for peer review.

TECHNICAL CORRECTIONS

Line 346: centred -> centered

Line 374 might be out of place, the preceding paragraph talks about the correlation between mean winds calculated by the VVP and traditional all-sky method, while the concluding sentence mentions a semidiurnal tide in the polar region.

Line 420: This sentence does not flow, suggest a grammar revision: "Qualitatively, the zonal eastward/westward winds are like to corresponding to the positive/negative the horizontal divergence values."

Line 422: "shows more complicate" -> "shows more complicated"

Line 425: "needs more explored" -> Does not flow.

Line 450: "increase in the meteor number" -> Does not flow.

**Response:** Thank you for the suggestion. We have checked for syntax errors and corrected those errors accordingly.

Comment: The Angle-of-Arrival calculations starting on Line 189 and following are well laid out. 1) Line 192ff: "AOA can be determined...": Include at least a reference or a brief description of the process, given that the bi-static case has unique challenges compared to the monostatic case. This will help substantiate the statement on Line 344" Through bistatic geometry, the coordinates of the meteor locations (x, y) can be deduced." 2) Line 196: "This small difference can be calculated...": Include an equation and/or a reference to the procedure. 3) Perhaps discuss any hardware-specific challenges encountered such a calibrating system phase biases.

**Response:** Thank you for your valuable suggestions. The Mengcheng-Changfeng bistatic meteor radar system is basically similar to the bistatic meteor radar systems described by Stober and Chau (2015) and Spargo et al., 2019. Therefore, methods for estimating the radial velocity, meteor position, angle of arrival, etc., mainly refer to their works. As the reviewer suggested, we have added the relevant references to the revised manuscript.

Comment: Vertical Components: Line 334: Discuss the validity of setting the vertical component, w0 = 0 especially in light of the addition information that multistatic meteor radar brings (e.g. multiple viewing angles). Such a simplification is typical in traditional all-sky meteor radar processing, but perhaps discuss if and why this continues to be a valid and appropriate approach.

**Response:** We have revised equations (7) – (8) and added the vertical velocity (as shown in Figure R1) to the calculation, taking the approach from Stober and Chau (2015) and Chau et al. (2017). The resulting divergence value decreases due to the recalculation. However, the resulting vertical velocity reaches a few m/s, which is more similar to the apparent vertical velocity due to horizontal divergence contamination (Chau et al., 2017). Vincent et al. (2019) estimated the polar vertical velocity using the divergence of the mean meridional wind using wind measurements of more than 20 years, and the resulting magnitude was several cm/s. Stober et al. (2022) introduced a 3DVAR+DIV algorithm to estimate the real vertical velocity, and they obtained vertical velocities in the range of $\pm 1$–2 m/s for most of the analysed data during 2 years of

collection. In the next stage, we will expand our multistatic meteor radar network in Central-Eastern China. The new multistatic meteor radar network will provide a better determination of vertical winds refer to the more recent results reported by Stober et al. (2021; 2022).

[Figure]

**Figure R1.** The vertical velocity of (a) October 16 and 17, 2021, and (b) February 1, 2022, to February 1, 2023. The thin black dashed lines in each panel present the equinoxes.

**Comment:** Line 425ff: any discussion on previous work, or specific opportunities for exploration over your geographic area?

**Response:** In the revised manuscript, to provide more new information based on the bistatic meteor radar system, we extended the observation data to more than 1 year and added a new section in the manuscript. This study will provide the first investigation of the seasonal variations in MLT dynamics at lower-midlatitudes (near 30°N) in the Northern Hemisphere.

We added estimations of the gravity wave (GW) momentum fluxes over a year, and our results are consistent with those of previous studies. For example, the seasonal variation in gravity wave (GW) momentum fluxes observed by the bistatic meteor radar system

shows good agreement with that of Jia et al. (2018). The zonal GW variances indicate a semiannual variation, with two maxima around April and October and two minima in August and December. Placke et al. (2011) reported that the GW activity in the upper mesosphere over Collm, Germany (51.31°N, 13.01°E) at higher-middle latitudes has a semiannual oscillation with the main maximum in summer and a minor maximum in winter.

Moreover, we added the seasonal variation in horizontal wind deformations in the MLT region, but in this respect, at present, there are no similar observations for intercomparison at these latitudes.

**Comment:** Line 363: "In order to verify the reliability of our results, we compared the traditional all-sky results and the VVP results by calculating the correlation coefficients and the regional winds" If the overall aim of the paper is to relax the homogeneity assumption, presumably to model the true wind field with better accuracy, is it appropriate to use the correlation between the traditional method (which is defined as making certain simplifying assumptions) and the VVP method which in theory should be more accurate? In other words, is the traditional method an appropriate 'ground truth' metric? In the extreme case, the traditional method's simplifying assumptions are invalid, in which case validating a new method based on high correlation with the traditional method is inappropriate.

**Response:** As the most important tools for measuring MLT winds, the results of meteor radar have been generally accepted and compared with those of many other measuring tools (Franke et al., 2005; Reid et al., 2018; Zeng et al., 2022; and the references therein). However, most meteor radars in the world operate in the form of a single station, and all use the traditional least square method to estimate the winds. Therefore, if the results of VVP are highly correlated with those of traditional methods, it can at least prove that our VVP method can present a result close to the value of the wind field of a single station. For the accuracy and reliability of the results, at least in the traditional method, the more meteors the results are, the more reliable, and our system can provide nearly

twice as much data as traditional single-site systems. In the future, we will utilize the Gaussian method (Stober et al, 2018; Hindley et al., 2022) to reverse and improve more accurate wind results.

**References in this response:**

Conte J F, Chau J L, Urco J M, Latteck R, Vierinen J, Salvador J O. 2021. First studies of mesosphere and lower thermosphere dynamics using a multistatic specular meteor radar network over southern Patagonia. *Earth and Space Science*, *8*, e2020EA001356. https://doi.org/10.1029/2020EA001356

Chau J L, Stober G, Hall C M, Tsutsumi M, Laskar F I, Hoffmann P. 2017. Polar mesospheric horizontal divergence and relative vorticity measurements using multiple specular meteor radars. *Radio Science*, *52*, 811–828. https://doi.org/10.1002/2016RS006225

Hindley, N.P., et al., Radar observations of winds, waves and tides in the mesosphere and lower thermosphere over South Georgia island (54° S, 36° W) and comparison with WACCM simulations. Atmospheric Chemistry and Physics, 2022. 22(14): p. 9435-9459

Jia M, Xue X., Gu S, Chen T, Ning B, Wu J, et al. 2018. Multiyear observations of gravity wave momentum fluxes in the midlatitude mesosphere and lower thermosphere region by meteor radar. *Journal of Geophysical Research: Space Physics*, *123*.https://doi.org/10.1029/2018JA025285

Placke, M., Stober, G., and Jacobi, C.: Gravity wave momentum fluxes in the MLT–Part I: seasonal variation at Collm (51.3° N, 13.0° E), J. Atmos. Sol.-Terr. Phy., 73, 904–910, 2011.

Reid, I.M.; McIntosh, D.L.; Murphy, D.J.; Vincent, R.A. Mesospheric radar wind comparisons at high and middle southern latitudes. Earth Planets Space 2018, 70, 84

Spargo A J, Reid I M, MacKinnon A D. 2019. Multistatic meteor radar observations of gravity-wave tidal interaction over southern Australia. *Atmospheric Measurement Techniques*, *12*(9), 4791–4812. https://doi.org/10.5194/amt-12-4791-2019

Stober, G., Liu, A., Kozlovsky, A., Qiao, Z., Kuchar, A., Jacobi, C., Meek, C., Janches, D., Liu, G., Tsutsumi, M., Gulbrandsen, N., Nozawa, S., Lester, M., Belova, E., Kero, J., and Mitchell, N. (2022) Meteor radar vertical wind observation biases and mathematical debiasing strategies including the 3DVAR+DIV algorithm, *Atmos. Meas. Tech.*, 15, 5769–5792, https://doi.org/10.5194/amt-15-5769-2022.

Zeng, J.; Yi, W.; Xue, X.; Reid, I.; Hao, X.; Li, N.; Chen, J.; Chen, T.; Dou, X. Comparison between the Mesospheric Winds Observed by Two Collocated Meteor Radars at Low Latitudes. Remote Sens. 2022, 14, 2354. https://doi.org/10.3390/rs14102354

**Tracked changes**

**Multistatic meteor radar observations of two-dimensional horizontal MLT wind**

Wen Yi [1,2], Jie Zeng[1,2], Xianghui Xue [1,2,3*], Iain Reid[4,5], Wei Zhong[1,2], Jianfei Wu[1,2], Tingdi Chen[1,2], Xiankang Dou[1,6]

[1] Deep Space Exploration Laboratory / School of Earth and Space Sciences, University of Science and Technology of China, Hefei 230026, China
[2] CAS Key Laboratory of Geospace Environment/CAS Center for Excellence in Comparative Planetology, Anhui Mengcheng Geophysics National Observation and Research Station, University of Science and Technology of China, Hefei, China
[3] Collaborative Innovation Center of Astronautical Science and Technology, Harbin, China
[4] ATRAD Pty Ltd., Adelaide, SA 5000, Australia
[5] School of Physical Sciences, University of Adelaide, Adelaide, SA 5000, Australia
[6] Electronic Information School, Wuhan University, Wuhan, China

**Corresponding author:** Xianghui Xue (xuexh@ustc.edu.cn)

**Abstract:** All-sky meteor radars have become a reliable and widely used tool to observe horizontal winds in the mesosphere and lower thermosphere (MLT) region. The horizontal winds estimated by conventional single-station radars are obtained after averaging all meteor detections based on the assumption of the homogeneity of the horizontal wind in the meteor detection area (approximately 200-300 km radius). In this study, to improve the horizontal winds, we apply a multistatic meteor radar system consisting of a monostatic meteor radar in Mengcheng (33.36 °N, 116.49 °E) and a bistatic remote receiver in Changfeng (31.98 °N, 117.22 °E), separated by approximately 167 km to increase the number of meteors by at least 70% and provide two different viewing angles of the meteor echoes. The accuracy of the horizontal wind measurement depends on the meteor number in time and altitude intervals. Compared to typical monostatic meteor radar, our approach shows the feasibility of estimating the two-dimensional horizontal wind field. The technique allows us to estimate the mean

horizontal wind and the gradient terms of the horizontal wind, as well as the horizontal divergence, relative vorticity, stretching and shearing deformation of the wind field. In addition, the seasonal variation in MLT dynamics parameters, including horizontal wind deformations and gravity wave (GW) momentum fluxes, in the MLT region at middle latitudes of the Northern Hemisphere are presented. 
[revised manuscript text omitted]
., 2004). The composite error estimations are shown in Figure 12 (lower four rows) utilizing the data from October 16 to October 20, 2021. It is clear that the errors of horizontal winds and gradient terms are smaller than 1 m/s and 0.1 m/s/km, respectively, when the meteor detections are sufficient, such as the results ranging from 82 to 94 km during local morning (2000-0004 UT). The large errors basically occur above 94 km during local night (near 1200 UT), which is mainly caused by a low number of meteors.

Our results verify the ability of the VPP method to estimate  wind parameters. Based on these parameters, multistatic meteor radars are capable of deducing the inhomogeneities and kinetic characteristics of the wind fields, which are similar to those of Stober and Chau (2015). The increased meteor detections can reduce the error of the estimated terms and guarantee the reliability of the results.

**5. Seasonal variations dynamics parameters in MLT region**

This section uses the methodology described in the previous section to estimate the seasonal variation in MLT dynamics parameters, including horizontal wind deformations and gravity wave (GW) momentum fluxes, in the MLT region in the middle latitudes of the Northern Hemisphere from February 2022 to February 2023. As shown in Figure 13, we present the seasonal variations in monthly mean horizontal winds, horizontal divergence, relative vorticity, stretching deformation, and shearing

deformation. In Figure 13a, the zonal component shows westward winds in spring (March, April and May) and eastward winds in summer, autumn and winter (May to February) below 84 km, which is the characteristic of annual variation. However, within 84 to 90 km, the zonal wind shows maxima value in summer and winter as well as minimum value at the spring and autumn equinox, suggesting semiannual variation. From 90 to 96 km, the zonal wind again shows annual variation, with large eastward winds in spring to summer and low westward winds in autumn to winter. The overall zonal wind between 80 and 96 km is characterized by annual variation. During spring to summer, the zonal wind increases (greater eastward and lesser westward) with altitude, reaching a peak value of approximately 50 m/s at 96 km in May. In autumn and winter, the zonal wind decreases (lesser eastward and greater westward) with altitude, reaching a minimum value at 96 km in November.

In Figure 13b, the meridional winds show similar structures to the zonal winds. Below 86 km, the meridional wind is southward in spring to summer and northward in autumn to winter. Meanwhile, at latitudes between 88 and 96 km, the meridional wind is northward in spring to summer and is northward in autumn to winter, and the maximum values of northward and southward winds are 20 m/s and 15 m/s in May and October, respectively.

[Figure]

**Figure 13.** The seasonal values of horizontal winds and wind deformations from February 1, 2022 to February 1, 2023 at altitudes ranging from 80 to 96 km: (a) mean zonal wind, (b) mean meridional wind, (c) horizontal divergence, (d) relative vorticity, (e) stretching deformation, and (f) shearing deformation.

In Figures 13c and 13d, the horizontal divergence and stretching deformation are calculated by the gradient terms $u_x$ and $v_y$, and they portray a similar seasonal variation. The divergence is almost negative below 84 km. Within 84 to 96 km, the value is still negative in March to June, reaching -0.1 m/s/km at 96 km in May. In the remaining months of the year, the divergence is positive, reaching 0.08 m/s/km at 94 km in November. The maximum values of divergence (convergence) correspond to the

maxima of eastward/northward (westward/southward) wind velocities. For the stretching deformation, the result is totally negative below 88 km and shows the most negative value of -0.08 m/s/km at 80 km in January. In the upper altitude, the results are nearly negative in April to August and nearly positive in September to February, reaching 0.05 m/s/km at 96 km in February, which also corresponds to the maxima of the horizontal winds.

The relative vorticity in Figure 13e shows a clear stratification with height in summer and autumn, which is similar to the polar results reported by Chau et al. (2017). The value is negative between 84 and 92 km, with a maximum of -0.08 m/s/km in August at 88 km, and is positive below or above this height range. The maximum positive value is 0.18 m/s/km in January at 96 km. Finally, the wind shearing deformation, as shown in Figure 13e, is close to 0 and even negative near the spring and autumn equinox, which corresponds exactly to the annual variation structure of the background mean winds. The minimum and maximum values are both at 88 km, -0.04 m/s/km and 0.09 m/s/km, respectively, in March and December.

[Figure]

**Figure 14.** Height-time cross-sections of the zonal (a) and meridional (b) wind variance and the vertical flux of zonal (c) and meridional (d) momentum. Running averages over 30 days shifted by 7 days from February 2022 to February 2023. The black dashed lines indicate the zero mean zonal wind in (a) and (c) and zero mean meridional wind in (b) and (d).

Compared to monostatic meteor radar, more meteor detections and different viewing angles of the radial velocity observed by the bistatic meteor radar system can appreciably improve the precision of the GW covariance estimation (Spargo et al., 2019). In addition, we also try to use the bistatic meteor radar system to estimate the GW momentum fluxes, basically following the method introduced by Hocking (2005), Jia et al. (2018), Spargo et al. (2019) and the references therein. Figure 14 presents the seasonal variations in GW momentum flux terms.

As shown in Figures 14a and 14b, the zonal ($u'^2$) and meridional ($v'^2$) GW variance terms generally increase with height below ~90 km and then decrease with height above 90 km, reflecting the characteristics of the GW amplitude variation. The zonal GW variances indicate a semiannual variation, with two maxima around April and October and two minima in August and December.

The meridional ($v'^2$) GW variance mainly shows annual variation, with a maximum amplitude in winter and spring and a minimum amplitude in summer and autumn. Below 90 km, most gravity waves are unsaturated, so the amplitude of the gravity wave increases exponentially due to the decreasing air density, thus causing the GW variance to increase with height. Up to approximately 90 km, when the gravity wave is saturated, according to linear theory, the gravity wave will become unstable and break, leading to a decrease in the GW variance (Fritts, 1984; Placke et al., 2011).

In Figures 14c and 14d, the vertical fluxes of zonal (v'w') and meridional (u'w') momentum basically show a U-shape and an inverted U-shape, respectively. Both u'w' and v'w' decrease with height throughout the year, except for v'w' in September, when both reach maxima in September and minima in June but at different corresponding altitudes. Moreover, the meridional component appears to be southward during almost the whole year. These results are generally comparable with those of a previous study observed by Wuhan (30.2°N, 114.2°E) and Beijing (40°N, 116.3°E) at middle latitudes (e.g., Jia et al., 2018). Placke et al. (2011) reported that the GW activity in the upper mesosphere over Collm, Germany (51.31°N, 13.01°E) at higher-middle latitudes has a semiannual oscillation with the main maximum in summer and a minor maximum in winter. Combined with the background wind structure, we can see the influence of gravity waves on the structure of the background mean winds. From May to August, it can be considered that only the westward gravity wave can propagate upward under the background of the prevailing eastward wind, and the meanwhile, the westward momentum flux will produce the northward drag force under the action of the Coriolis force and thus slow down the southward wind.

**6. Discussion and Summary**

In this study, we have presented  preliminary results from the Mengcheng and

Changfeng bistatic meteor radar systems. The main objectives were accomplished successfully by the new bistatic meteor radar system and are summarized as follows:

[revised manuscript text omitted]